# Time scales and gaps, Haar fluctuations and multifractal geochronologies

Shaun Lovejoy [1] ✉, Rhisiart Davies [1], Andrej Spiridonov[2], Raphael Hebert [3] & Fabrice Lambert [4]

Outcrops and cores are primary sources of information about the Earth's past. Quantitative analyses rely on geochronologies that take into account highly variable sedimentation and erosion rates as well as gaps from missing strata. Using 23 geochronologies from the Holocene, Quaternary, Phanerozoic and Precambrian, we apply Haar fluctuation analysis to statistically characterize the number of measurements per unit time - the measurement densities. The analysis determines the densities' (multifractal) scaling regimes and exponents; collectively, the analyses span over nine orders of magnitude in time scale. The measurement density is a new paleoindicator that we show is typically correlated with the primary paleoindicator, biasing and complicating its statistical interpretation. We also analyze the distribution of gaps linking the latter's (probability) scaling with series incompleteness and the length Sadler effect. The density characteristics are needed to unbias spectra and other statistical characterizations.

Much of our knowledge of Earth's past is gleaned from interpreting the stratigraphy of sedimentary, igneous, and metamorphic sequences, whether exposed in outcrop, recovered from cores through lake and ocean sediments, or preserved in crystalline crust. These were laid down under highly variable conditions, and their stratigraphy generally includes temporal gaps of all sizes[1]. Since erosion, sedimentation, aggradation, denudation and progradation rates of sediment sources and sinks operate over wide ranges of scale[2], the simplest hypothesis is that they are scaling in accord with scaling Holocene and Quaternary climate records[3–13] and analyses of Phanerozoic benthic stacks[14] and macroevolution[15] physically reflecting interconnections across various temporal scales and planetary geochemical reservoirs. In this study, we analyze chronologies and proxies from records spanning the Holocene, Quaternary, Phanerozoic and (much of) the Precambrian, focusing on the measurement density—i.e., the number of measurements per unit time—that we quantitatively characterize in a multifractal framework. This density is a new and distinct multi-scale proxy that can be used to understand the evolution of Earth's system history. Implicitly, it includes information on the absence of evidence, thus— paradoxically—providing positive information about the structure of non-recorded history, a significant phenomenon that can be quantified in its own right. We also analyze the scaling of the probability distributions of the inter-measurement time intervals and show that completeness of the records decreases with the number of measurements, thus quantitatively explaining the length Sadler effect[2,16–19].

Cores and outcrops are typically sampled regularly in depth; the resolutions are determined both by available technology and by their physical characteristics, such as the effects of bioturbation (various oceanic and terrestrial sediments), molecular diffusion (isotopes in ice cores), and material redistribution at the surface (e.g., by winds or currents). However, even when regularly sampled in depth, geochronology is typically non-uniform. Although adequate for identifying and dating strong transitions ("jumps"), or intense events ("spikes"), this can pose serious problems for statistical analysis (including for Lomb periodograms[20]).

Today, series with hundreds of thousands of measurements are increasingly available; therefore, statistical analysis is mandatory. How should we handle highly variable temporal resolution? Either one adapts the method to the data or the data to the method. An example of the former is Haar fluctuation analysis[21] (see below) that characterizes how fluctuation amplitudes change with time scale and can be conveniently implemented on nonuniform chronologies[14]. Alternatively, one may create a regular timeline, interpolate the data, and then use conventional techniques, such as Fast Fourier Transforms. Although convenient and commonly employed, interpolation can lead to uncontrolled biases[22]. The problem is that the data are typically "rough": fractal, multifractal, and nondifferentiable, whereas standard interpolation functions are smooth (differentiable of order 1 or more), which leads to biases[8,20].

Regardless of the analysis strategy, it behooves us to study, characterize, and understand the statistics of the chronologies themselves. This knowledge is required, if only to allow us to assess and statistically correct the

[1]Department of Physics, Montreal McGill University, Montreal, QC, Canada. [2]Department of Geology and Mineralogy, Faculty of Chemistry and Geosciences, Vilnius University, Vilnius, Lithuania. [3]Alfred-Wegener Institute Helmholtz Centre for Polar and Marine Research, Potsdam, Germany. [4]Geography Institute, Pontificia Universidad Católica de Chile, Santiago, Chile. ✉e-mail: shaun.lovejoy@mcgill.ca

biases. Therefore, initial characterizations of inhomogeneities should attempt to characterize the statistical properties of chronologies over a wide range of scales, which requires scaling techniques. For records with only a few hundred measurements, it may be sufficient to theorize inhomogeneity in terms of geometric fractal sets. If time is discretized at the finest resolution, then at any location on the time axis, there is a binary choice: either there is a measurement or none. Either the data or their complements (holes and gaps) may form a fractal set. Such a black/white binary reduction is the temporal analog of treating spatial distributions of meteorological measuring stations as a fractal set, which implies that the measurements have not only limited spatial but also limited *dimensional* resolutions[23,24]. However, with big data, a more complete characterization is needed, which requires us to follow the spatial example[25] that instead considers the fundamental quantity to be the *measurement density* $\rho(t)$ which is a mathematical *function* with a value at each point on the time axis (more precisely, if it is scaling, $\rho(t)$ is in fact the density of a singular multifractal measure—it is a "generalized function"). Figure 1 shows an example with 11,874 points, showing that $\rho(t)$ may have huge variability, displaying typical multifractal "spikes" (see the "spike plots" in ref. 26, see also Fig. S1). The relationship between the fractal set and multifractal density descriptions was clarified in ref. 27,28, and discussed in the climate context[29].

This paper is a collaboration of specialists in Holocene, Quaternary, Phanerozoic and Precambrian time scales as well as nonlinear geophysics, presenting an empirical study of geochronology measurement densities, discussed here over nine orders of magnitude in scale (Table 1 and SUPP). By analyzing $\rho(t)$, we show how chronologies—including the gaps—can be understood in a scaling multifractal framework. We also determine the scale-by-scale correlation of fluctuations $\Delta\rho(\Delta t)$ with fluctuations in the paleo indicators themselves ($\Delta T(\Delta t)$). If $\Delta\rho(\Delta t)$ and $\Delta T(\Delta t)$ are statistically independent, then statistically correcting the biases is easier. Alternatively, correlations or similarities in scaling may indicate some ontological causal connections between the time series and the cross-scale unevenness of their measurement densities.

The science of stratigraphy has recognized for a long time the significance of temporal gaps in diverse records and across time scales. In concert with the lithological and facies succession features, the genetic discontinuities between neighboring rock formations are used in identifying the presence of gaps in stratigraphical records, and thus defining sequences in a correlational framework of the sequence stratigraphy, which searches for the best ways to objectively distinguish space-time "packages" of the geological record[1]. A precursor to sequence stratigraphy—allostratigraphy—was and still is explicitly concerned with revealing the ranges of

discontinuities and gaps in the geological record, which are later used in correlation between distant areas[30]. Gaps appear at all time scales, and depending on their magnitude can be graded from diastems (relatively short interruptions on geological time scales) to disconformities in relatively continuous and lithologically homogenous successions, to angular unconformities which signify not only gaps in the record but also structural and tectonic deformations which happened in the unobserved time, and to complete nonconformities when sedimentary rocks or sediments are found to be overlaying much older igneous and metamorphic rocks, which usually signal gaps with durations reaching 100 Myrs to Gyrs, Therefore the classical stratigraphy already implicitly recognized the scaling and scale free nature of the gaps in the record and chronologies. The present study quantifies this structure of the gaps as a function of time scales over which they occur, while also exploring their correlations with proxies, which show clear scale-dependence of these correlations. The presence of correlations between proxies and the gaps in their records suggests two things: i) proxy record is systematically biased in non-trivial ways with the scale-dependent magnitudes of distortion; ii) there should be a common cause, implicit in both allo- and sequence stratigraphical approaches, which drives paleoclimatic signals and the formation of gaps in their records. Therefore, paraphrasing the famous aphorism: "the absence of evidence is also the evidence of absence". It means that the time scale-dependent (scaling) structure of gaps in records of proxies can be used as a new and mathematically tractable source of information in revealing the Earth system processes.

## Results
### Comparisons using the ensemble of data over scales from years to Giga years

We selected 22 paleoclimate series from individual cores covering the Holocene (past 12 kyr), upper and middle Quaternary (past 0.8 Myr). In addition, for the longer time scales, we chose a benthic stack (compiled from numerous ocean cores past 470 Myrs, Fig. 1) as well as a Precambrian $\delta^{18}$O carbonate (past 3.054 Gyrs, Fig. 3 and SUPP). These longer time scales were completed with a sea level series based on cores and outcrops[31] (Table 1, full analyses in the SUPP). Unlike the single-core data, these longer series were therefore able to fill some of the gaps, although many still remain.

To estimate the density $\rho(t)$, we extracted the chronologies and discretized the time axis into bins at the highest nominal data resolutions (see Fig. S2 and discussion), at which scale $\rho(t)$ is the (binary) indicator function (zero in bins with no data, 1 in bins with data). These densities were then analyzed using Haar fluctuation analysis[21] (based on the Haar wavelet). The Haar fluctuation of $\rho(t)$ at resolution $\Delta t$ is simply the difference between the average over the first and second halves of the interval: $\Delta\rho(\Delta t) = \overline{\rho_{[t,t-\Delta t/2]}} - \overline{\rho_{[t-\Delta t/2,t-\Delta t]}}$ (overbars for temporal averages, below - except for the correlations – we consider absolute fluctuations).

In a scaling regime, the mean fluctuation is a power law: $\langle\Delta\rho(\Delta t)\rangle \approx \Delta t^H$ where $H$ is the fluctuation exponent and "$\langle\rangle$" indicates ensemble averaging, here estimated by averaging over all the available disjoint intervals at scale $\Delta t$. $H$ characterizes how the mean fluctuations decay ($H < 0$) or grow ($H > 0$) with scale; it is one of an infinite (multifractal) hierarchy of exponents. However, the hierarchy itself can typically be characterized by only two additional exponents: the intermittency exponent $C_1$ and the multifractal index $\alpha$ (Methods). $C_1$ characterizes spikiness and its sharp transitions (e.g., Fig. 1); Gaussian processes are nonintermittent with $C_1 = 0$. $C_1$ was estimated as the logarithmic slope of the function $F(\Delta t)$ (see Methods), which is related to the ratio of the mean to the RMS fluctuation: $\log\left(\langle\Delta\rho(\Delta t)\rangle / \langle\Delta\rho(\Delta t)^2\rangle^{1/2}\right) = a\log F(\Delta t)$ where $\frac{1}{2} \leq a \leq 1$ is a weak function of $\alpha$ (Methods). Finally, the value of the multifractal index $\alpha$ is theoretically restricted[32]. to the range $0 \leq \alpha \leq 2$ when $H = 0$, $\alpha = 0$ corresponds to a binary monofractal "$\beta$ model", and $\alpha = 2$ to a pure "lognormal" multifractal.

The exponents may be physically interpreted using multifractal sedimentation and erosion models of $\rho(t)$. For these models, $H$ controls the basic scaling of the sedimentation rate, whereas the intermittency exponent $C_1$ controls sharp gradients, "spikes" in $\rho(t)$. Combined with $\alpha$, they also control

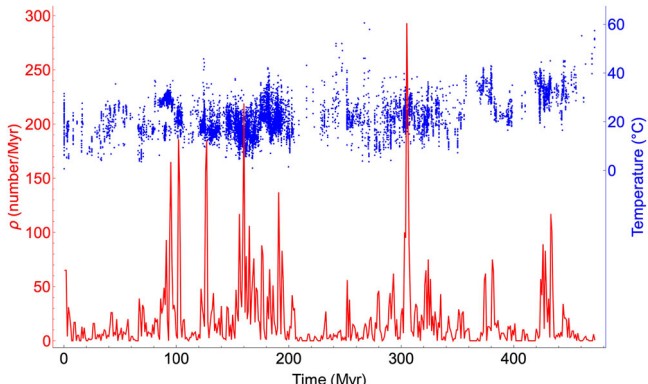

**Fig. 1 | Temperature data recorded in a paleotemperature reconstruction[56] (blue) and the corresponding measurement density as a function of time (red).** Large gaps in the data correspond to small measurement densities and vice versa. This dataset was a benthic stack at 10kyr nominal resolution, such that data from well-defined layers were used with many data points assigned to each 10kyr interval. There were 11874 data points for 3950 different 10kyr intervals over 470 Myr. The mean density was 25.3/Myr, see also Fig. 2 and S2.

**Table 1 | The datasets that were analysed in this investigation**

| Time Period | Data | Proxy type | Short name, Reference | Location | Time range | Resolution(nominal, in years) | Data points |
|---|---|---|---|---|---|---|---|
| | | | | (°N,°E) | | | |
| Holocene | Temperature | Speleothem | EurSpain[41] Oregon[42]2025-11-10 11:04:00 AM | (43, 4) (42.1, −123.41) | 3972–23 yr 8038–247 yr | 1 1 | 875 2678 |
| | | Pollen | Gonghai[43] HOB[43] Kettle[43], Moosent[43] | (38.9, 112.23) (47.69, 9.01) (48.61, −103.6) (47.02, 7.48) | 15–0 kyr 13–0 kyr 13–0 kyr 19–0 kyr | 6 6 6 6 | 777 860 551 513 |
| Quaternary | Dust | Ice Cores | EDC[44] Talos Dome[45], NGRIP RECAP[46] | (−75.1, 123.35) (−72.82, 159.18) (75, −42.3) (71.3, −26.72) | 801–0 kyr 150–4 kyr 108–10 kyr 121–5 kyr | 1 1 4 30 | 298205 103397 2774 2317 |
| | | Marine Sediment | Central PacificCentral South Pacific[47] | (4.68, -160.05) (-54.22, -125.43) | 141–7 kyr 474– 1 kyr | 60 40 | 289 2384 |
| | | Loess | Xifeng[48] | (35.7, 107.6) | 801–1 kyr | 200 | 722 |
| | Temperature | Ice Cores | Fuji Dome[49] EDC[50] EDML[51] WAIS[52], NEEM[53], 11/10/25 11:04:00 AM | (−77.3, 39.3) (−75.1, 123.35) (−75, 0.07) (−79, −112) (77.45, 51.06) | 340–0 kyr 801–0 kyr 150–1 kyrv 68–0 kyrv 129–2 kyr | 70 5 7 1 0.1 | 1189 5787 2303 6375 24434 |
| | | Lake | Tanganyika[54] | (−6.65, 29.8) | 59–1 kyr | 60 | 210 |
| | | Marine | Alkenone[54] | (37, −123) | 161–3 kyr | 300 | 284 |
| | Temperature Anomaly | Ice Cores | Vostok[55] | (75.1, −42.3) | 423–0 kyr | 9 | 3309 |
| Phanerozoic, Precambrian | Temperature | Benthic Stack | Grossman[56] | - | 470–0 Myr | 6 k | 11874 |
| | Sea Level | Marine Sediment | Haq Sea Level[31] | - | 128 Myr | 20 k | 617 |
| | δ[18]O Carbonates | Assemblage | Isson[33] | - | 3.504 Gyr-0 | 1 k* | 25399 |

The sample datasets used in the main body of the study were: Moosent, NEEM, South Pacific, EDC, Grossman, and Haq Sea Level. With the exception of the (long-time) benthic stack (Grossman), sea levels (Haq) and assemblage Isson (bottom rows), the shorter time scale series were all from single cores.

* The nominal resolution of the Isson assemblage is highly depend on the age, see Fig. S2; 1 kyr used in Fig. 3 is a compromise.

the statistics of the low ρ(t) regions (where there are long intervals between measurements). However, an additional explicit erosional model may be required to model these gaps, as shown below.

The results of the Haar analysis for normalized mean fluctuations $<\Delta\rho(\Delta t)>/\bar{\rho}$ are shown in Fig. 2, top. To simplify the figure, only representative datasets are shown; for the remaining series, see the SUPP. Despite often disparate origins (sediments, ice, and speleothems), the series shows two regimes. At higher frequencies, $H \approx -0.5$ and $C_1 \approx 0$ (the parameters of Gaussian white noise, note that mathematically, 1-D series must have $0 \leq C_1 \leq 1$). $H < 0$ implies that when averaged over progressively longer timescales, ρ(t) tends to converge to well-defined values. Maximum convergence—often at scales $\Delta t$ a hundred or more times larger than the finest resolution—occurs at a transition scale $\tau_c$ beyond which $H$ is positive, so that fluctuations now increase with timescale: the mean measurement density itself is no longer well-defined. At lower frequencies (Fig. 2 middle), we see that this new "wandering" ($H > 0$) regime (with $H$ typically $\approx 0.2$–0.3, Fig. 2 top) has $C_1 \gtrsim 0.1$ (Table 2). $C_1$ quantifies the rate at which intermittency builds up with scale, and the empirical values (often $\approx 0.2$) are substantial. In comparison, atmospheric turbulence typically[6] has $C_1 \approx 0.05$– 0.1, yet in spite of these apparently small values, at human scales, the atmosphere appears highly intermittent because the variability starts at planetary scales, ten million times larger. Finally (Table 2, SUPP), we find that the multifractal index α also transitions at $\Delta t \approx \tau_c$ (from $\alpha \approx 0$ to $\alpha \approx 1.0$ - 1.5).

Figure 2 also shows (gray line) the fluctuation analysis - including the qualitative $H < 0$ and $H > 0$ transitions - in paleotemperatures[14] $T(t)$. This figure demonstrates that the temperature and analogous ρ(t) regimes are qualitatively close; in any given regime, they have the same sign of $H$, such that $T(t)$ and ρ(t) are both convergent or divergent together.

## Correlations between measurements and measurement densities

Each proxy record has two potentially independent pieces of information: the paleoindicator $T(t)$ and the density at which it is estimated ρ(t). Figure 2 bottom, shows the normalized correlation coefficients ($R(\Delta t)$) between fluctuations $\Delta\rho(\Delta t)$ and $\Delta T(\Delta t)$. At small $\Delta t$, $R(\Delta t)$ of each dataset is typically low, indicating statistically independent processes. However, at $\Delta t > \tau_c$, correlations generally grow stronger, often roughly as $R(\Delta t) \propto \log \Delta t$ linear on the log-linear plots shown: (Fig. 2 bottom).

These correlations make the statistical unbiasing and interpretation of paleoindicators more difficult. As $R(\Delta t)$ increases, $\Delta\rho$ and $\Delta T$ tend to vary together, such that when both are large, paleoindicators tend to be very frequently sampled, and when both are small, paleoindicators tend to be rarely sampled. Both effects lead to biases. This situation arises, for example, in ice cores: the precipitation and hence sedimentation rate are temperature-related, implying $\Delta\rho(\Delta t)$-$\Delta T(\Delta t)$ correlations at these time scales (see e.g. the NEEM curve in Fig. 2 bottom). For most series, the sign of $R(\Delta t)$ was the

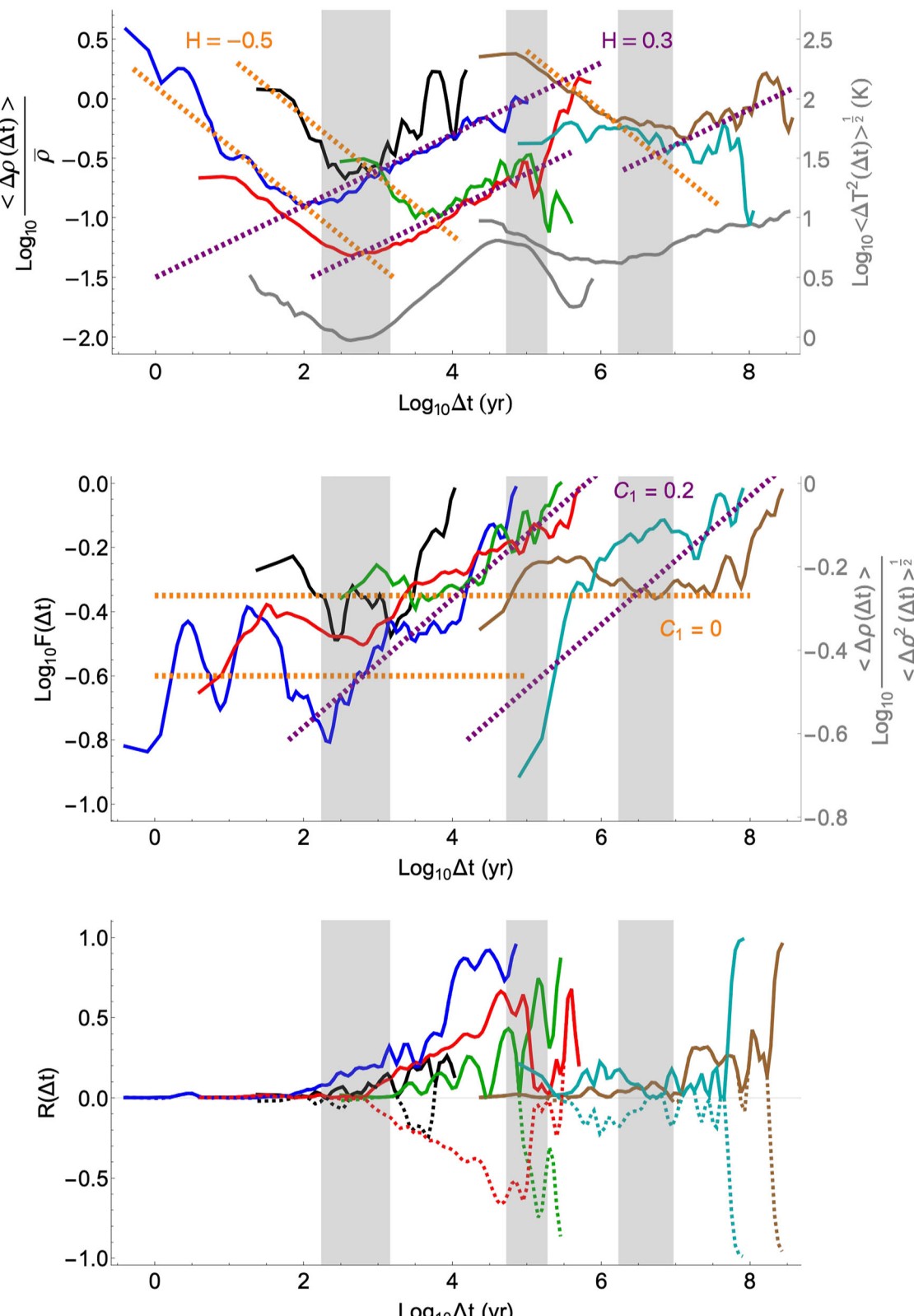

**Fig. 2 | Results of the Haar analysis of representative datasets.** Top: The normalized mean Haar fluctuation $<\Delta\rho(\Delta t)>$, the logarithmic slope of which gives an estimate of $H$. The two lower gray curves show the fluctuations in temperature for the EDC (left) and Grossman (right) datasets, showing that macroweather, climate, macroclimate, and megaclimate regimes are quite similar for $\rho$ and for $T$. Middle: plot of $\log_{10}F(\Delta t)$, as defined in the methods section, the slope of which gives an estimate for $C_1$. This plot differs from the log-log plot of the ratio between the fluctuation in measurement density and its root mean square by a constant coefficient $a$ for constant $\alpha$, which is shown for $\alpha = 1.3$ on the right axis. Bottom: the correlation coefficient between fluctuations in measurement density and fluctuations in the measured quantity. The solid lines are absolute correlations, and the dotted lines show negative correlations where relevant. The vertical gray regions in all three plots show the transition timescales between the scaling regimes in the climate system.

**Table 2 | Estimates for the critical timescale $\tau_C$, the fluctuation exponent $H$, the intermittency exponent $C_1$ and the multifractal index $\alpha$ for all datasets analysed**

| Time Period | Data | Proxy type | Short name, Reference | $\tau_C$ (yr) | $H$ $\Delta t<\tau_C$ | $H$ $\Delta t>\tau_C$ | $C_1$ $\Delta t<\tau_C$ | $C_1$ $\Delta t>\tau_C$ | $\alpha$ $\Delta t>\tau_C$ |
|---|---|---|---|---|---|---|---|---|---|
| Holocene | Temperature | Speleothem | EurSpain[41] | 60 | −0.5 | 0.3 | 0 | 0.1 | 1.1 |
| | | | Oregon[54] | 40 | −0.3 | 0.3 | 0 | 0.3 | 1.2 |
| | | Pollen | Gonghai[43] | 250 | −0.5 | 0.4 | 0.1 | 0.2 | 1.1 |
| | | | HOB[43] | 400 | −0.4 | 0.4 | 0 | 0.2 | 1.1 |
| | | | Kettle[43], Moosent[43] | 1600 | −0.2 | 0.3 | 0 | 0.3 | 1.1 |
| | | | | 400 | −0.8 | 0.4 | 0 | 0.2 | 1.0 |
| Quaternary | Dust | Ice Cores | EDC[44] | 400 | −0.5 | 0.3 | 0 | 0.1 | 1.5 |
| | | | Talos Dome[45], | 320 | −0.2 | 0.2 | 0 | 0.2 | 1.2 |
| | | | NGRIP | 5000 | −0.1 | 0.4 | 0 | 0.5 | 1.3 |
| | | | RECAP[46], | 6300 | 0.3 | −0.3 | 0 | 0.2 | 1.0 |
| | | Marine Sediment | Central Pacific Central South Pacific[47] | 4000 | −0.3 | 0.4 | 0.2 | 0.2 | 1.2 |
| | | | | 4000 | −0.5 | 0.4 | 0 | 0.2 | 1.2 |
| | | Loess | Xifeng[48] | 8000 | −0.3 | 0.3 | 0.1 | 0.1 | 1.3 |
| | Temperature | Ice Cores | Fuji Dome[49] | 16,000 | −0.5 | 0.5 | −0.3 | 0.5 | 0.8 |
| | | | EDC[50] | 2500 | −0.5 | 0.5 | 0 | 0.2 | 1.4 |
| | | | EDML[51] | 1600 | −0.5 | 0.6 | 0 | 0.1 | 1.5 |
| | | | WAIS[52], NEEM[53], | 400 | −0.5 | 0.6 | 0 | 0.1 | 1.4 |
| | | | | 110 | −0.5 | 0.3 | 0 | 0.3 | 1.5 |
| | | Lake | Tanganyika[54] | - | −0.1 | | 0.1 | | - |
| | | Marine | Alkenone[54] | - | −0.1 | | 0.1 | | - |
| | Temperature Anomaly | Ice Core | Vostok[55] | 3200 | −0.5 | 0.5 | 0 | 0.4 | 1.4 |
| Phanerozoic, Precambrian | Temperature | Benthic Stack | Grossman[56] | 12 M | −0.5 | 0.2 | 0 | 0.2 | 1.3 |
| | Sea Level | Marine sediment | Haq Sea Level[31] | 20 M | −0.2 | 0.2 | 0.1 | 0.2 | 1.3 |
| | $\delta^{18}O$ Carbonates (see the SUPP) | Assemblage | Isson[33] | 3–10 M | −0.1 | −0.05 | 0.2 | | 0.5–1.5 |

$H$ and $C_1$ estimates are for $\Delta t<\tau_C$ and $\Delta t>\tau_C$, whereas only $\alpha$ for $\Delta t>\tau_C$ is shown because for $\Delta t<\tau_C$, $\alpha$ transitions from the monofractal $\alpha \approx 0$ to a multifractal value. Neither the Tanganyika nor the Alkenone datasets undergo a scaling regime transition, and $\alpha$ is observed to increase across all timescales for both these datasets, hence, an estimate for $\alpha$ is not included for these two datasets.

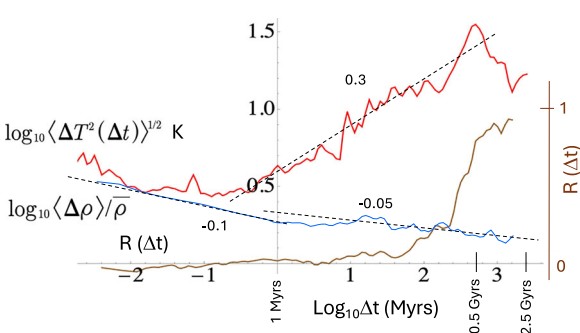

**Fig. 3 | A comparison of the root mean square (RMS) fluctuations of the Isson[33] $\delta^{18}O$ in Carbonates (red, roughly calibrated in K) with the normalized measurement density (blue) and the correlation between the two ($R(\Delt)$, brown, linear vertical scale at right).** There were 25399 data points, the oldest being 3.504 Gyrs, so that this fluctuation analysis takes us close to the origin of the Earth. As described in the text, the Isson $\delta^{18}O$ were roughly calibrated so that the results are expressed in units of $K$. The density fluctuations are dimensionless as is $R$. The dashed reference lines with slopes 0.3 and -0.05 are close to the RMS temperature fluctuations and normalized density fluctuations and cover the megaclimate ($\approx 0.5$ Myrs to $\approx 0.5$ Gyrs).

same at all $\Delta t$ values, and because the qualitative effects of $R \approx 1$ and $R \approx -1$ are the same, Fig. 2 bottom shows the absolute correlation (solid). However, in some cases, the sign of $R(\Delta t)$ did change with $\Delta t$, indicated by the dashed lines.

## A Precambrian example covering most of the Earth's history

Figure 2 was limited to Phanerozoic data, hence with a maximum $\Delta t$ of several hundred Myrs. Let us now briefly consider the Isson[33] $\delta^{18}O$ Carbonate series that is significant because it spans most of the Earth's history (3.045 Gyrs with 25399 measurements, Fig. S1). Being an assemblage from different authors, locations, it is much more heterogeneous than the other series considered here (for a discussion of some of the epoch specific issues, see ref. 34). For example, over the most recent 50 Myrs there are many segments with nominal resolutions of 10 kyr and even 100 yr (Fig. S3), yet the parts older than 600 Myrs (roughly the Phanerozoic), generally have resolutions of 1 Myr. This implies that the smaller $\Delta t$ statistics come exclusively from the Phanerozoic. Since these represent 81% of the measurements, to a good approximation, the statistics for $\Delta t \approx <600$ Myrs characterize only the Phanerozoic (see Figs. S1–7).

With these caveats in mind, Fig. 3 shows the RMS fluctuations of the $\delta^{18}O$ values (red, multiplied by 4, see below), $<\Delta\rho(\Delta t)>/\overline{\rho}$ (blue) and the correlation $R(\Delta t)$ (brown). Significantly, the $\delta^{18}O$ fluctuations are scaling over the entire megaclimate regime from $\approx 0.5$ Myrs $<\Delta t <\approx 0.5$ Gyrs, and the density fluctuations (slope −0.05) are also scaling over the same range. The sudden transition at $\Delta t \approx 0.5$ Gyrs agrees with the upper limit of the megaclimate inferred from analysis of the (biostrata-based) Geological Time Scale[29].

It is not clear how to interpret the larger $\Delta t$ decrease in variability; Fig. S4 shows that the Phanerozoic and Precambrian variabilities had the same exponent up until about 40 Myrs, but differ at the larger $\Delta t$, with the Precambrian having slowly decreasing variability at larger $\Delta t$. This indicates that these longest time scales are marginally stable in contrast to the $H > 0$ (unstable) Phanerozoic part. Even considering the largest $\Delta t$ (2.5 Gyrs corresponding to 2 (largely) disjoint intervals), this regime's range of scales

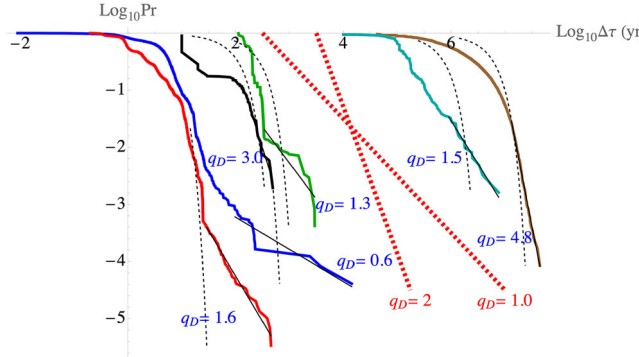

**Fig. 4 | The probability distribution $Pr(\tau' > \tau)$ of intervals $\tau$ between consecutive datapoints.** The black dotted lines show a fitted Gaussian distribution, whereas the straight black lines show the best-fit linear curve at the extremes of the data. The exponents $q_D$ are shown for the linear log-log fits, where $Pr(\tau' > \tau) \propto \tau^{-q_D}$. Dashed red reference lines have slopes $q_D = 1$ and $q_D = 2$ corresponding respectively to diverging means and variances. The NEEM data were particularly extreme with $q_D \approx 0.6$, and since the Gaussian fits were obtained from the means and standard deviations, in this case, the Gaussians are particularly far from the data. It is worth noting that the two Phanerozoic datasets were taken from multiple locations: the Grossman series is a benthic stack, and the Haq Sea Level curve was assembled from different outcrops in various basins. This effectively fills in holes such that they are more complete and have fewer gaps. See Fig. S7 for analysis of the Isson assemblage (Fig. 3) with $q_D \approx 1.5$.

(factor of $2.5/0.5 \approx 5$) is probably too short to be meaningfully interpreted as a new dynamical regime.

Returning to the megaclimate range, its exponent $H \approx 0.3$ is nearly identical to those from both the Veizer[35] and Zachos[36] benthic stack paleotemperatures (analyzed in ref. [14]), so that their statistics differ from the Isson series by a multiplicative constant. The paleotemperature stacks can thus be used to statistically calibrate the Isson assemblage. In Fig. 3, the latter were multiplied by 4 to be close to the Veizer paleotemperatures (for Zachos, the factor is $\approx 2$).

Considering the measurement densities, they show as an inflection at scales $\approx 3$ Myrs. While this is much less pronounced than for those in Fig. 2 (top), even flat ($H \approx 0$) $< \Delta\rho(\Delta t) > / \bar{\rho}$ curves imply quite nontrivial statistics: long-range statistical dependencies, and hierarchical clustering of density spikes (e.g., Figs. 1 and S1). We may also note that the density and the values are essentially uncorrelated until there is a fairly sharp rise in $R$ at roughly the age of the Phanerozoic ($\approx 0.5$ Gyrs), indicating that the interpretation of the longer $\Delta t$ statistics has some biases. Also note that since there are a few large $\Delta t$ segments, the correlations are high at the very extreme $\Delta t$'s (see the discussion in the SUPP).

### Measurement intervals, measurement gaps and the length Sadler effect

There is a well-known tendency for longer and longer paleo records to become less and less complete due to missing strata, we'll call this the length "Sadler effect"[16] to distinguish it from the original and related "resolution Sadler effect" in which completeness changes with the resolution which is controlled by a different exponent (see ref. [29]). To investigate this, we determined the probability of a random intermeasurement interval $\tau'$ exceeding a fixed $\tau$: $Pr(\tau' > \tau)$ (Fig. 4, S7 for Isson). We see that at small $\tau$, the distribution is (often) approximately Gaussian (dashed black lines). However, at large $\tau$, there is a transition to a scaling distribution $Pr(\tau' > \tau) \propto \tau^{-q_D}$, indicating a hierarchy of extreme intervals - gaps with small $q_D$ implying the existence of huge gaps. For example, for the NEEM, $q_D \leq 1$ so that the longest gap in a chronology is of the same order as the sum of all shorter gaps.

To see how the power law gaps explain the length Sadler effect, consider a series of $N$ measurements. If $q_D > 1$, the mean interval will converge so that

the overall length is $\tau_N \propto N$. The extreme gap will have the minimum probability $Pr_{min} \approx 1/N$, so that the extreme (maximum) gap length $\tau_{max} \propto Pr_{min}^{-1/q_D} \approx N^{1/q_D} \propto \tau_N^{1/q_D}$ – and therefore incompleteness (maximum gap) – grows with length $\tau_N$, hence the length Sadler effect (if $q_D \leq 1$, the effect is even stronger).

### Discussion and conclusions

Much of our knowledge of the evolution of life and the planet is inferred from stratigraphic records that are marred by hugely varying sedimentation rates and gaps where swaths of data are missing. The emergence of "big" paleodata makes it particularly necessary to characterize this variability, not only to statistically correct biases but also for understanding and modeling, including the underlying dynamical regimes (defined by the scaling properties) and that constitute the "continuum" variability. For this purpose, we exploit a new paleoindicator, the temporal measurement density $\rho(t)$. To isolate the statistics at different timescales $\Delta t$, we used Haar fluctuation analysis to statistically characterize fluctuations $\Delta\rho(\Delta t)$ from 24 paleoseries, collectively spanning a timescale range $\Delta t$ from years to 2.5 Gyr. Although the chronologies are products of both objective (and complex) physical processes as well as subjective technical data extraction issues, it is important to objectively determine their statistics, here, through their fluctuation statistics.

Analysis revealed that the series shared various scaling characteristics. Partial exceptions were the Haq paleo sea level and Isson $^{18}$O Carbonate assemblage, which used data from multiple locations to help fill in gaps. The series could be divided into distinct high- and low-frequency scaling regimes separated by a transition time scale $\tau_c$. In each regime, we quantified the statistics using the three basic multifractal parameters $H$, $C_1$, and $\alpha$. At high frequencies, $\Delta t \leq \tau_c$, we found $H \approx -0.5$, $C_1 \approx 0$, $\alpha \approx 0$ (Gaussian white noise), whereas at low frequencies $H \approx 0.2$–$0.3$, $C_1 \approx 0.1$–$0.25$, $\alpha \approx 1$–$1.5$ (Figs. 2 and S4). We note that in certain cases (notably ice cores), the layers are increasingly compacted with depth, so that the chronologies are not statistically homogeneous. However, as shown in the SUPP, this is a low-frequency effect that can be largely removed by polynomial detrending, so that this sediment compression does not impact the results of this study. The benthic stack assemblage and sea level series that combined data from multiple locations to help fill in gaps had similar qualitative behavior, but without evidence for a Gaussian regime.

Overall, four $\rho(t)$ regimes were identified (Fig. 2), which were in close agreement with the paleotemperature ($T(t)$) dynamical regimes identified in ref. [14] on the basis of the sign of $H$: macroweather (several weeks to centuries to millennia), climate (up to 50–100 kyr), macroclimate (up to 500 kyrs) and megaclimate (from $\approx 0.5$–3 Myr; up to $\approx 0.5$ Gyrs (Figs. 2, 3 and SUPP). Beyond $\Delta t \approx 0.5$ Gyrs, the Isson analysis showed that fluctuations in $T(t)$ transition from increasing to decreasing (stable) behavior (the $\rho(t)$ fluctuations are roughly constant). The general agreement between the $\rho(t)$ and $T(t)$ transition scales (and hence dynamical regimes) suggests that these have a deep significance, each corresponding to distinct nonlinear scaling processes governing both the climate system and erosion and sedimentation processes. While these scaling processes are presumably nonlinearly connected with each other, there is no implication that one *controls* the other.

$H < 0$ implies that as $\Delta t$ increases, $\rho(t)$ approaches a well-defined value: stable behavior, whereas at lower frequencies, $H > 0$, implying that fluctuations grow with scale, $\rho(t)$ seems to "wander", becoming strongly scale dependent. In this regime, the intermittency is strong ($C_1 \gtrsim 0.1$–$0.2$), so that $\rho(t)$ has strong transitions ("jumps", "spikes").

We also determined the scale-by-scale correlations $R(\Delta t)$ between $\Delta\rho(\Delta t)$ and $\Delta T(\Delta t)$, which are important for unbiasing the paleostatistics. It was found that in the $H < 0$ regime, $R(\Delta t)$ was generally small (statistical independence); however, for larger $\Delta t$ ($H > 0$), $R(\Delta t)$ increased roughly logarithmically with $\Delta t$.

At low $\rho$, the discrete nature of the chronologies is important, and it is necessary to quantify the gaps where the measurement interval $\tau$ is particularly large. For this, we determined the probability distribution of $\tau$, finding that there were typically abrupt transitions separating small and

large $\tau$, distinguishing "ordinary" intervals from those associated with missing strata that had power law probability "tails" with exponent $q_D$. These distributions imply that the length of the extreme gaps increases as $\tau_N^{1/q_D}$ where $\tau_N$ is the series length, implying that longer records have larger gaps and are less complete: the "length Sadler effect".

Our Haar fluctuation analysis directly handles inhomogeneous data (for periodic structures, see ref. 37) and has implications for statistical analyses. For example, if $<\Delta T> \propto \Delta t^H$, then $<\Delta T>/\Delta t \propto \Delta t^{H-1}$ so that the first derivative diverges when $H < 1$ (as we find here). More generally, $H$ is the highest order of (fractional) differentiation; therefore, linear ($H = 1$) or polynomial ($H \geq 2$) interpolations will be biased since the resulting interpolated series will combine some sections with $H < 1$ with others $H \geq 1$, and this, in an uncontrolled manner. Non-interpolative techniques such as Lomb-Scargle spectral analysis are generally strongly biased when applied to scaling data[22]. This is because scaling spectra are power laws: $\omega^{-\beta}$, and when $\beta > 0$, the spectra are dominated by low frequencies. Ignoring intermittency corrections $\beta = 1 + 2H$, so that problems arise whenever $H > -1/2$ (as found here). Other methods, such as the MultiTaper Method (MTM), are only theoretically justified for uniformly spaced data and are often unhelpful[20]. Future work is thus needed both to model $\rho(t)$ from more fundamental sedimentation and erosion processes, and to find optimum techniques for statistically correcting biases.

In the future, Haar fluctuation and trace moment analyses of the fossil record and the stratigraphical successions of environmental variables could be made in a wide variety of other systems, time periods and time scales. Ocean cores provide especially rich records of data and their gaps. In addition, the exploration of extremely deep time composite numerical records of hundreds of geological sections (e.g., Riedman and Sadler, 2018) could give new insights into the processes as well as biases, especially with respect to Precambrian biospheres.

In his seminal work "On the Origin of Species," Charles Darwin (1859) characterized the geological record as a book where "…history of the world imperfectly kept… entombed in our consecutive, but widely separated, formations". Here, invoking the power law scaling of extreme intervals (especially in the deep-time megaclimate regime), whose gaps become more extreme with longer records, we can add precision to his words: the gaps in paleo-proxy records become more pronounced at the longest time scales, where the most significant and dramatic changes in biota and the physical Earth become the most apparent.

Correlation analyses suggested a causal connection between the frequency of gaps and indicators of different Earth system parameters, such as surface or deep-water temperatures. This relationship can be seen as a bias in the estimation of numerical patterns (as it really is for many purposes), but also as an indication of the common cause, as was interpreted in the connection of the uneven formation of fossiliferous sedimentary packages and macroevolutionary processes[38]. Deep-sea records suggest that the amount of preserved sediments and all kinds of information recorded in them are dependent on climatic states[37]. In a sense, this is not surprising because a major branch of stratigraphy is based on the correlation of sedimentary packages bounded by sedimentary gaps—the sequence stratigraphy—which is implicitly based on the assumption of spatio-temporal coherency of climate effects (through the changes in the sea level and sedimentary facies) on the gaps in the sedimentary record[39]. Moreover, it was recognized that the gaps in the fossil record (and other stratigraphic records) can be seen not only as a bias of a quantity we want to characterize as accurately as possible, but also as a structure and feature worth exploration and explanation in its own right[40]. Therefore, the appearance of similar scaling regimes and correlations between paleoindicators and the densities of gaps in their records, but also the systemically changing nature of these correlations, reveals this new measure $\rho(t)$ as a new and largely unexplored indicator of past geodynamic, paleoclimatic, paleoecological, and evolutionary changes. At least two quantities need to be statistically characterized—the paleoproxy value itself, and also the temporal densities of values per unit time, which can provide additional and possibly non-redundant information about the paleogeography, and can be further analysed in numerous ways.

## Methods

Climate science has become increasingly specialized so that distinct (if overlapping) scientific communities specialize in the Holocene period (past 12 kyr), Quaternary epoch (past 2.6 Myrs) and Phanerozoic eon (past 540 Myrs), beyond the Precambrian. Each period, epoch, and eon exploits different paleoindicators, and each type of record has its own issues, problems, and limitations. The data types chosen here (details, Table 1, full analyses in the SUPP) are typical for the geological period covered, and the individual series were chosen for their quality and length, a total of 24. Despite the diversity, scaling analyses of their statistics (Figs. 2, 3, 4 and the figures in the SUPP) reveal that they can be placed in a common theoretical framework and, as summarized in Table 2, that they share basic statistical features over a wide range of scales.

### Analysis methods

**Multifractal processes.** We may represent geochronologies by the points on the time axis that delimit successive measurements, or—as here—by the temporal density of such points. The former representation is of a geometric set of points, the latter of a function $\rho(t)$. If the geochronology is scaling, then the points form a fractal set while $\rho(t)$ is the (singular) density of a multifractal measure. The relationship between the two descriptions was clarified in the context of deterministic chaos' strange (fractal) attractors[27,28]. This is discussed in the context of stratigraphy in ref. 29 as well as the reasons for preferring the multifractal description that we use throughout.

In scaling processes, fluctuations can be decomposed as:

$$\Delta\rho(\Delta t) = \varphi_\lambda \Delta t^H; \lambda = \frac{\tau_o}{\Delta t}; \Delta t \leq \tau_o \qquad \text{(M1)}$$

Where $\varphi_\lambda$ is the underlying intermittent driving process and $\tau_o$ is its outer scale. The statistics of $\varphi_\lambda$ may be defined by its $q$th order statistical moments: $\langle\varphi_\lambda^q\rangle = \lambda^{K(q)}$ with $K(1) = 0$ (corresponding to $<\varphi_\lambda> = 1$), so that $\Delta\rho$ is related to $\varphi_\lambda$ with the fluctuation exponent $H$. To characterize the fluctuation statistics at resolution $\Delta t$, we can determine the (generalized) $q$th order structure function $\langle\Delta\rho(\Delta t)^q\rangle$. Taking $q$th powers and averaging the above equation, we obtain:

$$\langle\Delta\rho(\Delta t)^q\rangle = \langle\varphi_\lambda^q\rangle\Delta t^{qH} = \tau_o^{K(q)}\Delta t^{\xi(q)}; \xi(q) = qH - K(q) \qquad \text{(M2)}$$

so that the structure function can be characterized by its exponent $\xi(q)$. Since $K(1) = 0$ we have $\xi(1) = H$.

The convex function $K(q)$ characterizes the multifractal, intermittent part, and due to the existence of stable, attractive multifractal processes[32] it is often assumed that $K(q)$ is of the "universal" form:

$$K(q) = \frac{C_1}{\alpha - 1}(q^\alpha - q) \qquad \text{(M3)}$$

So that $K(q)$ is determined by two fundamental parameters: the codimension of the mean $C_1$ and the intermittency index $0 \leq \alpha \leq 2$. From this we can verify that $K(1) = 0$, $K'(1) = C_1$, $\alpha = K''(1)/K'(1)$. A full characterization of the statistics of $\rho(t)$ over a given scaling regime is then determined by the triplet $H$, $C_1$, $\alpha$.

**The fluctuation exponent H.** As outlined above, $H$ characterizes the scaling of the mean ($q = 1$) with $H > 0$ indicating that fluctuations tend to grow with lag $\Delta t$ so that the series tends to "wander", and when $H < 0$, the fluctuations decrease with $\Delta t$, the fluctuations tend to cancel each other out.

**The intermittency exponent $C_1$.** The role of $C_1$ is less intuitive; for quasi-Gaussian processes, $C_1 = 0$ so that $K(q) = 0$ and only $H$ is important. In this special case and when $0 < H < 1$, the process is a fractional Brownian motion (with $H = 1/2$, it is classical Brownian motion), and when $-1 < H < 0$, the process is fractional Gaussian noise with the special case $H = -1/2$ corresponding to standard Gaussian white noise. $C_1 > 0$ characterizes the tendency for the series to make "jumps", transitions: "intermittency". For $\rho(t)$ this implies that there are occasional high density "spikes" (see Fig. 1) that are significantly larger than the mean, and these spikes form a fractal set with codimension $C_1$ (hence on the $d = 1$ time axis, a fractal dimension $1 - C_1$), as well as occasional long epochs where $\rho(t)$ is substantially below the average corresponding to long periods with few measurements. A more intuitive interpretation of $C_1$ was proposed in ref. 26 to characterize the intermittency of the basic dynamical regimes: it characterizes the rate of divergence of the ratio of mean to root mean square (RMS) fluctuation:

$$\langle \Delta\rho(\Delta t)\rangle / \langle \Delta\rho(\Delta t)^2\rangle^{1/2} = \left(\frac{\Delta t}{\tau_o}\right)^{aC_1} \quad (M4)$$

where (from Eq. 3 above), we find $a = \frac{(2^\alpha - 2)}{\alpha - 1}$ and since $0 \le \alpha \le 2$, the constant satisfies $\frac{1}{2} < a < 1$ and depends only (weakly) on $\alpha$ (in the chronologies, we found $\alpha \approx 0$ for $\Delta t < \tau_o$, $\alpha \approx 1.3$ ($a \approx 0.77$) for $\Delta t > \tau_o$, see Fig. S4).

**The multifractal index α.** The interpretation of $\alpha$—variously called "the multifractal index", "Levy index of the generator", and "degree of multifractality"—is more subtle. When $H = 0$, $\alpha = 0$, $C_1 > 0$, the process is the turbulent on/off "beta model", the process is non-zero over a fractal set with codimension $C_1$. When $\alpha = 2$ (the maximum), it is a "log-normal" multifractal with fluctuations following log-normal distributions (except for the extremes that have power law probabilities).

**Haar fluctuations.** Classical fluctuations are differences: $\Delta\rho(\Delta t)_{dif} = \rho(t) - \rho(t - \Delta t)$, and these can easily be estimated from irregular chronologies (e.g., pairwise differences). Another commonly used fluctuation is the "anomaly fluctuation" that is given by averaging segments over durations $\Delta t$ of series that had their overall means ($\bar\rho$) removed: $\Delta\rho(\Delta t)_{anom} = \overline{\rho_{[t, t-\Delta t]}} - \bar\rho$ where the overbar indicates the average over the interval indicated in the subscript. These common fluctuations have major—but complementary— shortcomings: for physical processes that decorrelate in time, mean difference fluctuations cannot decrease with lag $\Delta t$, whereas, on the contrary, mean anomaly fluctuations cannot increase with $\Delta t$. Haar fluctuations (based on the Haar wavelet) overcome both these limitations by combining averaging with differencing. The Haar fluctuation of $\rho(t)$ at resolution $\Delta t$ is simply the difference between the average over the first half and the second half of the interval:

$\Delta\rho(\Delta t) = \overline{\rho_{[t, t-\Delta t/2]}} - \overline{\rho_{[t-\Delta t/2, t-\Delta t]}}$. Like the difference fluctuation, it can easily be estimated from irregular chronologies (see methods and ref. 14, appendix B). For scaling processes where the average fluctuation varies as: $<\Delta\rho(\Delta t)> \approx \Delta t^H$, differences are adequate when $0 < H < 1$ and anomalies are adequate when $-1 < H < 0$. In comparison, Haar fluctuations are useful over the combined range $-1 < H < 1$ and this includes most Earth Sciences processes (for Gaussian processes, this corresponds to spectral exponents $\beta$ in the range $-1 < \beta < 3$). Beyond its ease of application and its broad generality, Haar fluctuations are also relatively easy to interpret, being nearly equal to either the difference or anomaly fluctuations (depending on the sign of $H$).

Haar fluctuation analysis was carried out on each dataset in Table 1. Empirical series have finite resolutions. However, when the chronologies are nonuniform, two slightly different strategies may be used to estimate the fluctuations. The first is outlined in ref. 14 here, we used a slightly different method, necessary because we are interested in the correlations between measurement density and the measured values. We therefore simply discretized the time axis at the finest resolution of the data and produced a uniform resolution series with values $T_i$ or a 0 according to whether or not there was data at the (uniformly spaced) time interval. For the Haar fluctuation of the data values between $t$ and $t - \Delta t$, we then took averages of the data values (i.e., excluding zeroes). The approximation being that where there were zeroes in the data, the values were close to the mean of the others in the segment duration $\Delta t/2$. For Haar fluctuations of the measurement density, we included the zeroes in the average. Any fluctuation involving segments with no data at all was rejected from the statistics. In numerous cases, we checked that this method gave very close estimates of the statistics as the previous method (we checked using various statistical moments as functions of scale). We also checked the process with various multifractal simulations of regular data with scaling "holes" with various statistics.

Finally, the fluctuations were multiplied by a "canonical" factor of 2 to make the Haar fluctuation closer to the difference fluctuation when $H > 0$ and closer to anomaly fluctuations when $H < 0$, see ref. 8, for the theory behind this. Note that when calculating the statistical moments, we averaged the absolute values of the fluctuations to the $q^{th}$ power over all available disjoint intervals.

**The intermittency: estimating $C_1$, $F(\Delta t)$.** The fluctuations in measurement density were normalized with respect to the mean measurement density over each dataset, $\bar\rho$, for comparison purposes. Using this simplified model, $\xi(1) = H$, therefore $\frac{\langle \Delta\rho(\Delta t)\rangle}{\bar\rho} \sim \Delta t^H$. The slope of the graph of $\log \frac{\langle \Delta\rho(\Delta t)\rangle}{\bar\rho}$ against $\log \Delta t$ yields an estimate for $H$ (Fig. 2 top). $C_1$ can be estimated from the function[6] $F(\Delta t)$, defined as:

$$\log F(\Delta t) = \log\langle\Delta\rho\rangle - \frac{\langle\Delta\rho \log \Delta\rho\rangle}{\langle\Delta\rho\rangle} \quad (M5)$$

In a scaling regime:

$$F(\Delta t) = \left(\frac{\Delta t}{\tau}\right)^{C_1} \quad (M6)$$

So that plotting $\log(F(\Delta t))$ against $\log(\Delta t)$ yields an estimate of $C_1$ as the slope of the graph (Fig. 2 middle). Combined with the relation for the mean to RMS ratio (eq. 4), we obtain:

$$a \log F(\Delta T) = \log\left[\langle\Delta\rho(\Delta t)\rangle / \langle\Delta\rho(\Delta t)^2\rangle^{\frac{1}{2}}\right] \quad (M7)$$

The plot of $\log(F(\Delta t))$ against $\log(\Delta t)$ therefore differs from the plot of $\log[\langle\Delta\rho(\Delta t)\rangle / \langle\Delta\rho(\Delta t)^2\rangle^{\frac{1}{2}}]$ against $\log(\Delta t)$ by a constant factor $a$. Taken in Fig. 2 middle to be approximately constant with value $a = 0.83$ corresponding to $\alpha = 1.3$ (close to the empirical value, see Fig. S4).

**Correlations between fluctuations.** If the dynamical regimes governing the $T(t)$ and $\rho(t)$ processes are the same, then it is possible that the $\Delta T(t)$ and $\Delta\rho(t)$ fluctuations are strongly correlated. At each lag $\Delta t$, we therefore estimate the correlation coefficient $R(\Delta t)$. This was done using:

$$R(\Delta t) = \frac{\langle\Delta\rho(\Delta t)\Delta T(\Delta t)\rangle}{\langle\Delta\rho^2(\Delta t)\rangle^{\frac{1}{2}}\langle\Delta T^2(\Delta t)\rangle^{\frac{1}{2}}} \quad (M8)$$

where $R$ is the correlation coefficient at a given timescale, $\rho$ is the measurement density, and $T$ is the measured quantity. This was done to explore potential systematic biases in the datasets. Note that the fluctuations are independent of any additive constant. However, strong trends could lead to nonzero mean fluctuations if present; these should be removed in the preprocessing (e.g., overall linear or quadratic trends, see Fig. S30).

If $\rho(t)$ is statistically independent of $T(t)$ (here, paleotemperature or dust flux), then high $\rho(t)$ is not associated with particularly large or particularly small $T(t)$ fluctuations, so that chronology corrections are easier. In

contrast, strong correlations ($R(\Delta t) \approx 1$) imply that $\Delta\rho(\Delta t)$ and $\Delta T(\Delta t)$ vary together. Intervals with large $\Delta\rho(\Delta t)$ have frequent measurements, so that $R(\Delta t) \approx 1$ implies that this tends to occur when fluctuations in $\Delta T(\Delta t)$ are also large. Conversely, there are relatively few measurements (very negative $\Delta\rho(\Delta t)$) when the fluctuations also have particularly low values (negative $\Delta T(\Delta t)$). When $R$ is close to 1, in intervals $\Delta t$ where $\Delta T$ is large, $T(t)$ is sampled very often, and conversely, in intervals where $\Delta T$ has low values, $T(t)$ is rarely sampled. Reversing the sign of $T(t)$ changes the signs of the fluctuations (it swaps "low" and "high" values of $\Delta T (\Delta t)$), and it changes the sign of $R(\Delta t)$, so that strong negative correlations $R(\Delta t) \approx -1$ are analogous to $R(\Delta t) \approx 1$ but for low values of the indicator.

**Time interval probability distributions and the length-Sadler effect.** While typical inter-measurement intervals depend on sedimentation processes and rates, individual cores typically also have "gaps" where by —as a consequence of erosion—whole swaths of sediment have completely disappeared from the record. Irrespective of whether or not a time interval is a consequence of a sedimentary or erosion process, at time $t$, the typical interval between successive measurements will be $1/\rho(t)$. Therefore, when $\rho(t)$ is small enough, $\rho(t)$ can no longer be treated as a continuous stochastic process. Mathematically, it suffices to instead treat the measurement process as a compound Poisson process controlled by $\rho(t)$, which is then a multifractal subordinator[29]. This model provides realistic simulations of the Geological Time Scale—i.e., the spacings of the "golden spikes" that define the main geological epochs, eras etc[29]. Full models of scaling sedimentation that follow the observed statistics are still needed.

These modeling issues are important, but are outside our present scope, which is the empirical statistical characterization of the measurement process. For this, it is enough to determine the probability of a random interval $\tau'$ exceeding a fixed $\tau$: $\Pr(\tau' > \tau)$ ( = 1–the (usual) Cumulative Distribution Function, CDF). Although at small $\tau$, the distribution is (often) roughly Gaussian (Fig. 4, dashed black lines, figures in the SUPP). However, at large $\tau$, there is a transition to a scaling distribution $\Pr(\tau' > \tau) \approx \tau^{-q_D}$, ("power law tails"), indicating a hierarchy of extreme intervals - gaps. Small values of $q_D$ imply the existence of huge gaps; for example, a value $q_D \leq 1$ (e.g., the NEEM record) implies that the duration of the longest gap in a chronology is of the same order as the sum of all the smaller gaps.

## Data availability
We produced no new data. The analyses were from the publicly accessible data sets that are detailed in the references cited in Tables 1, 2.

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

## Acknowledgements
We acknowledge the editors of *Nature*; *Nature Geoscienc*e; *Earth and Planetary Science Letters*; *Global and Planetary Change*; *Paleogeography, Paleoclimatology, and Paleoecology*. S.L. was supported by the National Science and Engineering Research Council (Canada) grant 222858. A.S. was supported by the project S-MIP-24-62 BretEvoGeneralized. F.L. was funded by ANID– Fondecyt 1231682.

## Author contributions
R.D. carried out data analysis for an undergraduate project and, along with S.L., interpreted the results. S.L. designed the project and supervised its completion. R.H., F.L., and A.S. provided datasets to analyse and contributed to the interpretation of the results. R.D. and S.L. wrote the manuscript with contributions from all authors.

## Competing interests
The authors declare no competing interests.
