## [Transparent Peer Review file · Communications Earth & Environment]

Time scales and gaps, Haar Fluctuations and multifractal Geochronologies

Corresponding Author: Professor Shaun Lovejoy

Version 0:

Decision Letter:

Dear Professor Lovejoy,

I apologize for our delay in sending this decision letter.

Your manuscript titled "Scaling Geochronologies" has now been seen by 3 reviewers, whose comments are appended below. In the light of their advice we regret to inform you that we cannot publish your manuscript in Communications Earth & Environment.

You will see that while some reviewers are interested in your study, all reviewers raise substantive concerns regarding the flaws in the assumptions made for the statistical analysis and flaws in providing additional robust evidence to support your conclusion about climatic control in the geological record. Reviewers also highlight the need to provide a more robust discussion that connects the conceptual framework with the natural record and the need to extrapolate your analyses beyond the Phanerozoic. Taking these points together with our editorial considerations, we are unable to conclude that your manuscript represents a sufficiently robust and compelling advance over the body of related work in the literature. Unfortunately, these reservations are sufficiently important to preclude publication of this study in Communications Earth & Environment.

We are sorry that we cannot be more positive on this occasion and thank you for the opportunity to consider your work.

please bear in mind that we are committed to providing a fair and robust review process. Please do not hesitate to contact us if you would like to discuss our decision.

Best regards,

Carolina Ortiz Guerrero, Ph.D.
Associate Editor,
Communications Earth & Environment
Consulting Editor, Communications Sustainability

Reviewers' comments:

Reviewer #1 (Remarks to the Author):

SUMMARY

The manuscript introduces an important, under-appreciated consequence of the incompleteness of the stratigraphic record: measurement density varies much more notably along geologic timelines than might be anticipated from the sample spacing on a thickness scale. This aspect deserves a wide audience and would be the primary reason to publish this manuscript. I recommend returning it for substantial revision, however, because I worry about two potential audiences. Those who struggle with the mathematical foundation of stratigraphic incompleteness are likely to get lost; I know from workshops and courses that the fractal properties easily become confusing. In contrast, readers who are very familiar with the consequences of stratigraphic incompleteness are likely to be frustrated by ambiguities and vagueness in the statements about measurements and scales. The manuscript refers to data sets that range from single, local, measured sections to what must

be composite sections compiled from multiple local sections. For the latter, how is the underlying correlation problem overcome? It adds another set of uncertainties and assumptions that seem to have been ignored.

The stratigraphic record is a much more reliable indicator of the sequence of events than about their temporal spacing. Even sequence, however, is locally incomplete. Correlating local sections into a more complete, age-calibrated record is fraught with uncertainties and assumptions. The manuscript does not deal effectively with this aspect of the data the authors have chosen.

STRENGTHS

The manuscript deals in detail with an under-appreciated property of stratigraphic time series. Measurement density of the data encountered in local stratigraphic sections and regional composite sequences is rendered much more variable and intermittent than the apparent regularity of sampling. This problem arises for two reasons. First, the physical record is inevitably highly incomplete on a wide spectrum of scales. Second, the correlation of multiple sections, which is intended to fill-in the local gaps, introduces more questionable, simplifying assumptions. This manuscript deals with the first of these problems. It suggests statistical remedies that will be new to many readers and the paper could become a valuable reference.

WEAKNESSES

The two title words do not provide good guidance for readers to anticipate the contents of the paper. Something like "Multiscale Measurement Densities in Geologic Time Series" might serve better.

The dependence of average rate upon the time span of measurement (the Sadler Effect) is a simple and inevitable mathematical outcome for intermittent and reversible time series like sediment accumulation. Yet, it confounds simplistic approaches to geologic time series. Linear interpolation, for example, will most often be ill-advised when assessing the duration represented by the thickness of an interval of a sequence of sedimentary layers. In my experience, it is not an easy concept for geological audiences to appreciate fully. At very short time scales, intermittency results from the particulate nature of sediment. On the scale of decades to millennia, intermittency and reversibility arise from the dynamics of sediment transport, climate change, and sea-level change. The longer-term accommodation of accumulating sediment depends more on the dynamics of tectonic subsidence. This spectrum of causes is well established in publications and should already be so very familiar to geoscientists that it is not advisable to simplify it.

For current purposes, the most important data in a sedimentary sequence are those that indicate the age of a layer. When this manuscript refers to data, it fails to indicate whether, or to what extent, the data points are indications of age or duration. These data allow a sequence to be calibrated as a time scale. Such data are critical missing information in Table 1, for example, which lists resolution without a sense of the source of the estimate. When the authors refer to statistics of chronologies, do they somehow know the age of all data points, or do they refer only to dated events. Is there an implicit interpolation of age, in spite of all the warnings about this at the heart of the subject? Is a published age-axis taken at face-value? Is the so-called nominal resolution merely what is written on the cited authors' axes?

Does the lack of geological clarity, originate because the native English-speaking author is not a geoscientist and the geologic expertise resides among co-authors who are not native English-speakers. The set of authors is impressive and surely brings all the necessary expertise to this subject.

DETAILS

Line 11 . . . knowledge of the Earth's past . . . Delete "the"

Line 11 . . . interpreting successive layers . . .

Line 12 ". . . sheets of rock . . ." I am unsure what is imagined.

Line 25 . . . thus quantitatively explaining the Sadler effect." Since the term was coined, the Sadler Effect has been discussed in more than 160 papers concerned with a wide range of natural processes. Its mathematical background was provided in several papers prior to this, notably by statistician David Strauss, co-authoring with Sadler, but also by others. Line 33 I agree that temporal resolution is highly variable. How, then, should we understand the single values in the 7th column in table 1? That table appears to give no information about the number of age control points. For speleothems, age control may be provided by counting annual rings but is also provided by Uranium-Thorium systematics. There is also the confounding factor of the relatively narrow range of the heights of the speleothems that have been dated.

Line 46 ". . . if time is discretized at the finest resolution . . ." Sediment is particulate. Would not the finest resolution be the arrival of individual particles? It may be worth considering the limiting time scale of the instantaneous arrival of individual sediment grains. At this limit, the accumulation rate tends to infinity and the stratigraphic completeness to zero. Although this might seem beyond practical value, it is another way to appreciate the inescapable logic of the Sadler Effect.

Line 47 Perhaps there are three options at any location on the time axis: either there is sediment/rock of that age or none and, if there is sediment/rock, there may be a measurement or none. The time and the thickness axes need careful separation. As the paper makes clear, elsewhere, a regular sampling interval in thickness is likely to be quite irregular in time. Reference to thickness and to time scales needs to be more carefully distinguished. Reference to time scales needs to be warily justified.

Line 51 Delete "ref"

Line 55-57 Delete "in ref". The manuscript uses three formats for citations: a superscript number alone, a superscript number preceded by "in" and a superscript number preceded by "in ref." The first format dominates and would seem appropriate everywhere.

Line 59 ". . . geochronology measurement densities . . ." Geochronology is the "study of time in relationship to the history of the Earth" (AGI Glossary of Geology). Here, as in the title, it seems simpler and more appropriate to use the word "time

series". The current wording leaves a geologist wondering whether all the measurements might be dated events. Given the large numbers of measurements reported, we later wonder whether any of the data points are age calibration events.

Line 70 What is the "10 kyr nominal resolution"? Does "nominal" refer to the original scaling ticks on the age axis? Does it refer to the age bins into which the author compiled data from many sources, with their own uncertainties. Table 1 lists a 6 kyr resolution for this data set, not 10 kyr. The number of data points allows only 1 data point per approximately 40 kyr, on average, across the 473 Myr time span. Perhaps this is enough that the authors can determine where I became confused.

Line 73 Do these paleoclimate series comprise proxy data measurements or interpretations?

Line 76 "We . . . discretized the time axis into bins at the highest (nominal) data resolution." Was this highest resolution only locally realized or was it the highest resolution that could be applied throughout the series. Did it rely on the original authors' scale markings or did it consider the available dated/datable events that that enabled age-calibration. Was interpolation of ages used? How?

Line 94 "H controls the basic scaling of the sedimentation rate." Although not in the form of the Haar function, Sadler's 1981 and later papers all describe incompleteness and patterns of unsteadiness in terms of the values and inflexion points of the slope of logarithmic graphs of average rate against time span.

Line 169-70 The Haq et al. Curve was indeed "assembled" from different outcrops and basins. It should matter to this manuscript how the data were "assembled". Does sequence stratigraphy assume a knowledge of the causes and durations of sequence, their bounding gaps and their "orders," which might compromise the use of the data-set here (line 230).

Line 207 A missing space after "as"?

Line 208 Only long series have the opportunity to span across a long gap, assuming these are gaps in rock/sediment (What geologists often term a hiatus.). This is one of the trivially obvious aspects of the Sadler Effect. For a measured time interval to include a hiatus, that interval must begin and end in rock (or sediment) that can be dated before and after the hiatus; it must be longer than the duration of the hiatus.

Line 220-1 There is a risk that this statement might imply that the more significant and dramatic changes must take more time. The geologic record surely shows that some dramatic extinctions were rather rapid. Of course, in order to appreciate "dramatic" we need a longer record of normal, background rates of change. Of course, longer measurement series improve the chance of capturing "rare" events.

Line 248 Double commas; delete one.

Line 251 Column justification fails in columns 5 and 6 of table 1.

Lines 251-2 How does the stated resolving power (7th column in Table 1) compare with the analytical uncertainty in the dating method and reasonable estimates of incompleteness? How many of the data points are determinations of age? The numbers of data points (8th column) rarely amounts to enough for 1 point per bin at the state resolution. What am I missing?

Line 265-6 This text implies that the data are available on a time axis, that temporal density is known. But the geologic time axes are not known with such resolving power, especially if linear interpolation between dated events is discredited.

Line 270 Reference to a paper that is in-preparation does not assist the reader. Delete "ref."

Line 396-297 Gaps result from both non-deposition and erosion. There need not have been "whole swaths of sediment that have completely disappeared from the record." There are times and places where none were deposited. Neither should the reader be thinking only of large gaps.

Reviewer #2 (Remarks to the Author):

This is the kind of study that many geologists have trouble with: data, including large data sets, being analyzed entirely divorced from their origins. Two observations about statistical analysis are pertinent: 1) It is possible to perform sophisticated statistical analysis data sets as though patterns in numbers alone can reveal significant truths. 2) An elementary requirement for meaningful statistical analysis is to ensure that the sample is representative of the population to be studied. Yes, we are told that the samples come from ice cores, or deep-marine sedimentary records, but there is no context to the individual records. There is no evidence that any attempt has been made here to explore either of the above two points.

The thing about geological data is that it is often very messy. We are talking about the stratigraphic record here, so there are all kinds of reasons why sampling may be incomplete: 1) the sampled item, such as a specific type of fossil, is limited by its ecology to only certain types of rocks, representing specific types of sedimentary environment. The occurrence of these represent at least two uncontrolled variables; 2) the rocks to be sampled are present/absent in response to any kind of geological variable – subsidence or uplift of the sampled basin, climate, selective erosion reflecting the rocks characteristics, etc. 3) Sampling may simply reflect availability, which is never amenable to regular spaced collection except from continuous drill cores. How many of your source data bases are of this type?

On lines 26-31 the authors acknowledge reasons why geochronologies might be non-uniform because of a range of uncontrolled variables, and then in the next paragraph it seems to be assumed that large data bases then lend themselves to meaningful statistical analysis of this irregularity, proposing that the statistical signals that emerge then have some geological meaning.

Just to take the very large data set used in the construction of Figure 1. Sample density is calculated and displayed graphically as if it has some geological significance. An examination of the paper from which this data set was obtained (reference 29) reveals all kinds of reasons for variable sample density: the data consist of several types of fossil, each constrained by its own ecological parameters; the samples were obtained from both carbonate and phosphatic sediments, which occur in different depositional settings; some of the intervals within which sample spacing is large represent time intervals of different climate or tectonic setting. It is hard to reason why sample spacing, in and of itself, would have any significant meaning at all.

The thrust of the paper seems to be to explain sample variability entirely as some sort of product of climate change. On line 228 reference is made to a single study (reference 31) that identified the long 2.4 my orbital cycle, and the authors seem to extrapolate from that to a broad assumption of climatic cyclicity everywhere. This particular referenced study (which I looked at), based on solid geological research, might well be correct. The 2.4 my cycle is well known in the geological record. But this is not proof that the entire study under review here and all the data used therein reflect climatic control on different time scales. In fact the orbital signal is not often clearly preserved in the sedimentary record because its influence is often quite subtle, and is swamped by other non-systematic geological processes, notably tectonic controls and the variability of sedimentary environments and processes. Please see my book "The origin of stratigraphic sequences" for reference.

The best things I can do is suggest that the authors read

Miall, A. D., 2015, Updating uniformitarianism: stratigraphy as just a set of "frozen accidents", in Smith, D. G., Bailey, R., J., Burgess, P., and Fraser, A., eds., *Strata and time: Geological Society, London, Special Publication 404*, p. 11-36.

This paper is attached to my review. The purpose of the 2015 paper was to examine the Sadler effect from the point of view of sedimentology, and to discover what actually drives the geological controls behind rates and styles of stratigraphic preservation at different time scales. Sample spacing was not a variable examined. And there is no statistics in the paper. The raw field data speaks for itself. It reveals a range of non-periodic geological processes acting simultaneously over a wide range of time scales. The only statistically regular geological control is astronomical, in the form of orbital forcing and tidal processes, and this is one of the major processes for sedimentary successions that can be dated as being formed within the tens to hundreds of thousands of years time range. Most geological sections are accumulated over periods of millions, or many millions of years, and geological processes acting over those time scales (including long-term climate controls) are episodic or irregular in time and rate.

My suggestion: send this paper to a full-time earth scientist who has complementary skills in statistical analysis.

Reviewer #3 (Remarks to the Author):

Review of Scaling Geochronologies

I like the general concept that there is information encoded in the lack of stratigraphic record as it imparts a constraint on the temporal significance of unconformities (or other processes blurring or erasing the record). Basically, this study suggests that measurement density itself, normally considered just a methodological limitation, could act as a new kind of proxy for understanding geological processes. This is a neat idea, and I commend the authors for a nicely written text that was generally a joy to read. I do have some comments which I hope are useful for the authors as they enhance the work. I look forward to seeing this work in print. Well done!

While we expect that older records have more missing data (this is intuitively self-evident), the key contribution here as I see it is; A statistical model that predicts the growth of missing intervals based on real data. A scaling law that describes how the probability of missing data changes over time. A recognition that correcting these biases when reconstructing past environments may be important.

More significant thoughts.

The concepts here are so important hence it is unfortunate that there are no examples from anything older than the Phanerozoic. I realise this is a moderately large ask but the use of this work would be vastly enhanced with something from the Precambrian. The Phanerozoic is only 12% of Earth's history, and given the work is about sampling density at the grandest conceptual scale it would seem reasonable to have a Precambrian example (of which I can think of plenty to explore).

Many readers would likely appreciate a stronger connection between the conceptual framework and the rock record.

Introducing a brief discussion on the diversity of unconformities, perhaps with a specific example or two, would enhance the text and provide greater geological context. A sentence or two in the introduction could effectively ground the discussion in observable field relationships, reinforcing the relevance of these concepts to real-world stratigraphic and tectonic settings.

Line 77 is important, and it is unlikely that anyone but specialists will get "is the (binary) indicator function of the boundary". Specifically, the concept that the binning is done at the highest possible resolution meaning the smallest available time step in the dataset needs more explicitly discussed. This process effectively transforms the temporal distribution of measurements into a sequence of ones and zeros, much like a presence-absence function. points to consider.

Line 12; the word stratigraphy would be useful here as applied to rocks, sediments, and ice. There of course is also the deep time record retained within magmatic systems and their inherited mineral cargo which is also not immune to the same data density concepts.

Line 13; insert the word time, so its time gaps.

Line 16; I would urge a little caution here as while sedimentary rocks evidently are driven by climate in part, there is also the reasoned counter argument that the rock cycle itself influences atmospheric cycles. Some concept of the interconnection across various temporal scales of the various planetary geochemical reservoirs would be nice to acknowledge.

Line 18; what is the difference between measurement density and number of measurements per unit time. If they are the same, as I presume, they are, then no need to repeat this statement as it serves to confuse not clarify.

Lines 29-40; The Lomb periodogram for unevenly sampled data see Press et al. 1992 probably deserves a mention.

Lines 39-40; This sentence really needs to be unpacked to help the readership, many of whom will not be time series experts but rather practitioners.

Line 49; To be honest the purpose of mentioning meteorological stations in respect to what you are describing is lost on me,

surely there is a more relevant example given the papers focus.

Line 77; I suspect mentioning the classic Haar wavelet, moving window, first before getting to the fluctuation would help the readership.

Line 20; a bit pedantic but for essentially all Earth Science measurements it is five POTENTIALLY independent pieces of information; the paleoindicator, the uncertainty on the paleoindicator measurement, the data density, the time indicator, the uncertainty on the time measurement indicator. While these factors can be analysed separately, they are often interdependent, especially when measurement gaps or dating uncertainties influence how we interpret paleoindicators.

Line 240; "infinite" seems a little of an over sell.

It would strengthen the paper to explore potential underlying physical processes that govern the scaling regimes discussed, as this could provide deeper insights into why measurement densities follow the observed patterns. Additionally, I find the argument compelling that assuming missing data is random leads to biased interpretations, and that recognizing a power-law distribution in missing data allows for statistical adjustments in reconstructions. It would be valuable to elaborate further on how such adjustments could be practically implemented (whether through resampling techniques, weighting schemes, or other statistical corrections) to guide future applications of this approach in paleoclimate and geochronological studies.

Chris Kirkland, Perth, WA

Version 1:

Decision Letter:

Dear Professor Lovejoy,

Your revised manuscript titled "Time scales and gaps: Haar Fluctuations and multifractal Geochronologies" has now been seen by 3 reviewers, and we include their comments at the end of this message. You will see that although the reviewers appreciate the effort you put into the revisions, reviewer 2 continues to raise concerns about the robustness of your analysis, and therefore the robustness of your conclusions. We are interested in the possibility of publishing your study in Communications Earth & Environment, but would like to consider your responses to these concerns and assess a revised manuscript before we make a final decision on publication. Specifically, for publication in Communications Earth & Environment to be appropriate, a revised manuscript must make a compelling case that your approach is robust and not simply a reflection of local availability of sample material.

We therefore invite you to revise and resubmit your manuscript, along with a point-by-point response that takes into account the points raised. Please highlight all changes in the manuscript text file.

Please submit your point-by-point responses as a separate file, distinct from your cover letter where you can add responses to the Editors' comments that you do not want to be made available to the reviewers. Word files are preferred. We recommend that any figures, tables or graphs that are included in the response to reviewers are also included in the main article or Supplementary Information.

Please use the following link to submit your revised manuscript, point-by-point response to the referees' comments (which should be in a separate document to any cover letter), a tracked-changes version of the manuscript (as a PDF file) and the completed checklist:

Link Redacted

We hope to receive your revised paper within six weeks; please let us know if you aren't able to submit it within this time so that we can discuss how best to proceed. If we don't hear from you, and the revision process takes significantly longer, we may close your file. In this event, we will still be happy to reconsider your paper at a later date, as long as nothing similar has been accepted for publication at Communications Earth & Environment or published elsewhere in the meantime.

Please do not hesitate to contact us if you have any questions or would like to discuss these revisions further. We look forward to seeing the revised manuscript and thank you for the opportunity to review your work.

Best regards,

Alireza Bahadori, PhD

Senior Editor
Communications Earth & Environment
Consulting Editor
Communications Sustainability

EDITORIAL POLICIES AND FORMATTING

- Behavioural and social science
- Ecological, evolutionary & environmental sciences
- Life sciences

Furthermore, please align your manuscript with our format requirements, which are summarized on the following checklist: <https://www.nature.com/documents/commsj-phys-style-formatting-checklist-article.pdf> Communications Earth & Environment formatting checklist

and also in our style and formatting guide <https://www.nature.com/documents/commsj-phys-style-formatting-guide-accept.pdf> Communications Earth & Environment formatting guide .

***** DATA:** Communications Earth & Environment endorses the principles of the Enabling FAIR data project (<http://www.copdess.org/enabling-fair-data-project/>). We ask authors to make the data that support their conclusions available in permanent, publically accessible data repositories. (Please contact the editor if you are unable to make your data available).

All Communications Earth & Environment manuscripts must include a section titled "Data Availability" at the end of the Methods section or main text (if no Methods). More information on this policy, is available at <http://www.nature.com/authors/policies/data/data-availability-statements-data-citations.pdf>.

If a community resource is unavailable, data can be submitted to generalist repositories such as <https://figshare.com/> or <http://datadryad.org/> Dryad Digital Repository. Please provide a unique identifier for the data (for example a DOI or a permanent URL) in the data availability statement, if possible. If the repository does not provide identifiers, we encourage authors to supply the search terms that will return the data. For data that have been obtained from publically available sources, please provide a URL and the specific data product name in the data availability statement. Data with a DOI should be further cited in the methods reference section.

REVIEWER COMMENTS:

Reviewer #1 (Remarks to the Author):

I remain very much in favor of accepting this manuscript for publication. The statistical Haar-fluctuation method that it presents deserves to be much better known among geologists and paleontologists. I look forward to using that approach and this paper would be my cited source.

The authors' revisions have added considerable clarity to the prior draft and title. I agree with them that little would be added by extending the data deeper into Precambrian time, the multi-fractal nature of the record is well-known.

I sympathize with the authors' evident frustration at some naivety that the earlier draft exposed in the reviewers' understanding, including mine. We reviewers are, however, likely to have brought more expertise to the topic than the average reader. The author's revisions are surely worthwhile for the overall impact of the publication, even at the cost of lengthening the paper.

I do remain somewhat confused by the intrinsic "nominal resolution" the authors ascribe to their data sources. I would not insist on any revision for this, because understanding their method does not seem to hinge on this value. Perhaps it would be useful, however, if I give an example that might explain my confusion. Consider Figure 12.5 (chapter 12, figure 5) in Elsevier's "A Geologic Time Scale 2004" edited by Gradstein, Ogg and Smith. The figure displays the interpolation of an age-scale through 22 radio-isotopically dated stratigraphic levels into an ordinal sequence of early Paleozoic events spanning almost 80 million years.

The spacing of the 22 dated levels displays the kind of irregularity (whether on the age scale or the composite, ordinal event scale) that the current manuscript will help geologists manage. The resolution of the individual dates is +/- 1.5 to 1.7 million years. The time-ticks on the age axis are 2 million years apart. On the other axis, the 22 dated events are imbedded in an ordinal sequence of the first and last appearance events of 669 fossil graptolite species and subspecies, together with other events (e.g. isotopic anomaly boundaries), for a total of 1,400 event levels. This ordinal composite sequence of events had been optimized, using simulated annealing, from the local records of 119 stratigraphic sections, each with its own unique pattern of gaps and missing events.

By the GTS 2020 edition of the Geologic Time Scale book (this one edited by Gradstein, Ogg, Schmitz and Ogg) the composite ordinal scale had grown to be based on 2,600 Ordovician to Devonian graptolite species range-end events, culled from 840 local stratigraphic sections for a total of 34,000 local events correlated and sequenced. A second composite sequence for 2,200 Cambrian to Silurian conodont species known from 1,300 sections was based upon a total of 41,000 local event records. Only about 150 stratigraphic sections in this global database yielded both graptolite and conodont species ranges for about 7,500 local range-end records.

Which of all the counts in the previous two paragraphs would be the intrinsic nominal resolution for these data? Although I am unsure, I doubt that has undermined my grasp of the Haar statistics.

Reviewer #2 (Remarks to the Author):

I see no discernible response to my comments on the first version of this paper.

I attempt to contact Dr. Lovejoy to discuss my issues informally, but have not had a response. As I explained in that communication, with my reasons, I cannot convince myself that geochronology sample spacing has any other meaning than the local availability of sample material. Accordingly I find the subject of the ms misleading, and cannot recommend its publication.

Reviewer #3 (Remarks to the Author):

Review of Lovejoy et al

I remain positive about this contribution, as I was in the previous round. The following minor but constructive points may help the authors further strengthen the paper.

Minor points

"Much of our knowledge of the Earth's past is gleaned by interpreting varying stratigraphy of layers in outcrops, cores drilled through lake and ocean sediments, and sheets of rock and ice." For a geologist this statement is technically a bit limited, in that layers in outcrop implies a metasedimentary pile what about all the information from the igneous rock record? I don't disagree with the statement at all, just it really would be more accurate for the subject to be framed as "Much of our knowledge of Earth's past is gleaned from interpreting the stratigraphy of sedimentary, igneous, and metamorphic sequences, whether exposed in outcrop, recovered from cores through lake and ocean sediments, or preserved in crystalline crust."

There is a structural issue in logic and temporal framing around line 14. The first sentence sets up a deep-time or all-Earth-history perspective, while the second abruptly narrows to Holocene–Quaternary dynamics without signalling the temporal zoom. To resolve this, the bridge between the two should explicitly define that the case studies or processes examined are focused on the recent geological past, even though the framework could apply more broadly.

Line 33 and 34 "the dated geochronologies are typically non-uniform", agree with the concept but the word geochronology is a tautology. The phrase "geochronology is typically non-uniform" does the job and consistent with community usage of what geochronology is. By definition, geochronology already refers to the dated temporal framework of geological materials or events. Adding "dated" is redundant.

Line 37; Personally, I don't massively like the zoom in to cores specifically, this work is equally as relevant to outcrops, mineral grains or mass spec analyses. As a geochronologist this framing may lose some readership in my community. I mention this only to be helpful.

Line 46; nice!

Line 63-65; Not sure you really need this claim of precedence?

Line 180; "Our knowledge of the evolution of life and the planet is largely inferred from stratigraphic records that are marred

by hugely varying sedimentation rates and gaps where swaths of data are missing” Debatable, this overstates the role of stratigraphic records in reconstructing the entire history of life and the planet. Beyond the Phanerozoic, the sedimentary archive becomes increasingly incomplete, while the igneous and metamorphic record, particularly through mineral chronometers and geochemical proxies, becomes the dominant repository of information.

Line 221; Please unpack this sentence a bit more “Our Haar fluctuation analysis directly handles inhomogeneous data (for periodic structures see ref), and has implications for statistical analyses. For example, H is the highest order of differentiation, typically <1 ; therefore, linear (or polynomial) interpolations ($H \geq 1$) will be biased”. This would benefit from being unpacked into two or three sentences that clarify both what the Haar fluctuation analysis does and why it matters for interpolation or scaling.

Line 231-233; this is probably more standard phrasing for the community; “the gaps in paleo-proxy records become more pronounced at the longest time scales, where the most significant and dramatic changes in biota and the physical Earth become the most apparent.”

I will highlight in this response not for something here, but maybe in the future. “The concepts here are so important hence it is unfortunate that there are no examples from anything older than the Phanerozoic. I realise this is a moderately large ask but the use of this work would be vastly enhanced with something from the Precambrian. The Phanerozoic is only 12% of Earth's history, and given the work is about sampling density at the grandest conceptual scale it would seem reasonable to have a Precambrian example (of which I can think of plenty to explore). Authors: “We totally agree that analyzing series longer than the Phanerozoic would be important. However in order to make convincing analyses, series with at least a hundred or so events are needed, and at the moment this seems to be lacking in the deeper geological record” Reviewers comment: It is not lacking in the deeper geological record. May be reach out to me, there are tons of truly deep time records that are sufficiently rich to try this type of approach on.

I'll also reiterate my earlier point, which may be helpful here to consider for a final time. As implied by Reviewer 2, broadening the paper's accessibility to a wider geological readership, not only those with a strong statistical background, would be achieved by providing a little more grounding in the rock record. While I appreciate that this is not the central focus of the study, including a few key statements about what the “missing record” looks like geologically would strengthen the conceptual foundation of the work, as this is its most fundamental premise. Even a short addition to the Introduction could effectively ground the discussion in observable field relationships, reinforcing the relevance of these concepts to real-world stratigraphic and tectonic settings. This could be achieved for example, through a few sentences on diverse unconformity types, aiming to frame observable field relationships to the statistical concept, enhancing accessibility for the widest geological audience.

This is a strong and interesting paper, and I offer these comments in the hope that they help strengthen its impact. Sincerely, Chris Kirkland

** Visit Nature Portfolio's author and referees' website at www.nature.com/authors for information about policies, services and author benefits**

Communications Earth & Environment is committed to improving transparency in authorship. As part of our efforts in this direction, we are now requesting that all authors identified as ‘corresponding author’ create and link their Open Researcher and Contributor Identifier (ORCID) with their account on the Manuscript Tracking System prior to acceptance. ORCID helps the scientific community achieve unambiguous attribution of all scholarly contributions. You can create and link your ORCID from the home page of the Manuscript Tracking System by clicking on ‘Modify my Springer Nature account’ and following the instructions in the link below. Please also inform all co-authors that they can add their ORCIDs to their accounts and that they must do so prior to acceptance.

Version 2:

Decision Letter:

Dear Professor Lovejoy,

Your revised manuscript titled "Time scales and gaps: Haar Fluctuations and multifractal Geochronologies" has now been seen by our reviewers, whose comments appear below. In light of their advice we are delighted to say that we are happy, in principle, to publish a suitably revised version in Communications Earth & Environment.

We therefore invite you to revise your paper one last time to comply with our format requirements and to maximise the

accessibility and therefore the impact of your work.

EDITORIAL REQUESTS:

****Please take care to match our formatting and policy requirements. We will check revised manuscript and return manuscripts that do not comply. Such requests will lead to delays. ****

SUBMISSION INFORMATION:

OPEN ACCESS:

Communications Earth & Environment is a fully open access journal. Articles are made freely accessible on publication. For further information about article processing charges, open access funding, and advice and support from Nature Portfolio, please visit <https://www.nature.com/commsenv/open-access>

Link Redacted

Best regards,

Alireza Bahadori, PhD
Senior Editor
Communications Earth & Environment
Consulting Editor
Communications Sustainability

REVIEWERS' COMMENTS:

Reviewer #1 (Remarks to the Author):

I recommend publication without further revision.

This is my third review of Lovejoy et al., the second for Springer-Nature. Each one was even more rewarding than the previous read. I remain convinced that this account of Haar-fluctuation statistics will be a valuable resource for all stratigraphers. It is novel, substantive, and surely ready for publication. It deserves a wide readership.

As before, the authors have responded well to the reviewers' suggestions. In particular, they have reacted positively to our misunderstandings by enhancing the clarity of crucial passages. As requested by other reviewers, the latest revisions extend the analyses back into pre-Phanerozoic time, where time series are longer, calibration is more difficult and gaps are likely to include some that are larger in duration and area. This is a bold undertaking, but valuable for readers. For the current version the authors have added seminal accounts of possibly fundamental differences between our knowledge of Phanerozoic and older time intervals. These indicate that the method they advocate might be able to indicate radical changes in the Earth system, in addition to inevitable scaling effects.

The Haar statistics are surely suitable for all forms of stratigraphic time series. The records of individual outcrops and cores,

as well as time series that have been subjectively compiled from many local sources, using the local estimates of age to build longer and richer composite series than any one source provides. The manuscript includes examples of both.

In addition, there are two particularly rich types of stratigraphic time series that deserve mention. First, there are the sea-bed drill-cores of the IODP, which are examined by collaborative, on-board and post-cruise teams of expert scientists, who use multiple dating and analytical techniques on the same material. Second are the studies that deliberately attempt to reduce the proportion of gaps in regional and global records. Whether for calibration of the Geologic Time Scale or for accounts of macroevolution, they apply objective sequencing algorithms to compile composite records from hundreds of published core and outcrop descriptions. There is now at least one example of the latter approach for Precambrian rocks (L. A. Riedman and P. M. Sadler, 2018, *Precambrian Research*, v. 318, p.6-18.)

Reviewer #3 (Remarks to the Author):

Thank you for your email. I have looked at the revision for COMMSENV-24-3266B. I have commented on this work several times now. This is the 3rd time. The authors have attempted to fully address all reviewer concerns and have improved the work. I remain positive about this work and recommend acceptance. I see no need for further rounds of review.

** Visit Nature Portfolio's author and referees' website at www.nature.com/authors for information about policies, services and author benefits**

Dear Professor Lovejoy,

I apologize for our delay in sending this decision letter.

Your manuscript titled "Scaling Geochronologies" has now been seen by 3 reviewers, whose comments are appended below. In the light of their advice we regret to inform you that we cannot publish your manuscript in *Communications Earth & Environment*.

You will see that while some reviewers are interested in your study, all reviewers raise substantive concerns regarding the flaws in the assumptions made for the statistical analysis and flaws in providing additional robust evidence to support your conclusion about climatic control in the geological record. Reviewers also highlight the need to provide a more robust discussion that connects the conceptual framework with the natural record and the need to extrapolate your analyses beyond the Phanerozoic. Taking these points together with our editorial considerations, we are unable to conclude that your manuscript represents a sufficiently robust and compelling advance over the body of related work in the literature. Unfortunately, these reservations are sufficiently important to preclude publication of this study in *Communications Earth & Environment*.

We are sorry that we cannot be more positive on this occasion and thank you for the opportunity to consider your work.

please bear in mind that we are committed to providing a fair and robust review process. Please do not hesitate to contact us if you would like to discuss our decision.

Best regards,

Carolina Ortiz Guerrero, Ph.D.
Associate Editor,
Communications Earth & Environment
Consulting Editor, *Communications Sustainability*

Dear Dr. Guerrero;

Thank you for sending the paper out to review and for giving your own appreciation. We deliberately held off resubmitting the paper because we decided to first publish the companion paper [Lovejoy *et al.*, 2025] "From Eons to Epochs: multifractal Geological Time and the Compound Multifractal - Poisson Process" in *EPSL*, and this has already garnered significant media attention. The *EPSL* paper uses the same statistical analysis techniques but applied to the Geological Time Scale and it shows how the latter respects an underlying scale invariance symmetry. This new submission is little more than an empirical analysis of measurement densities from 23 paleo records (the first time that this has been done) and the companion paper makes the analyses and interpretations more compelling.

Below, we give detailed responses to the referees, two of which (referee 1 and 3) were extremely positive. For example, referee 1 states:

"The manuscript introduces an important, under-appreciated consequence of the incompleteness of the stratigraphic record: measurement density varies much more notably along geologic timelines than might be anticipated from the sample spacing on a thickness scale. This aspect deserves a wide audience and would be the primary reason to publish this manuscript."

While referee 3 opines:

"I like the general concept that there is information encoded in the lack of stratigraphic record as it imparts a constraint on the temporal significance of unconformities (or other processes blurring or erasing the record). Basically, this study suggests that measurement density itself, normally considered just a methodological limitation, could act as a new kind of proxy for understanding geological processes. This is a neat idea, and I commend the authors for a nicely written text that was generally a joy to read. I do have some comments which I hope are useful for the authors as they enhance the work. I look forward to seeing this work in print. Well done!"

In my career, I have published over 200 papers and have rarely had such eulogistic comments from referees! In particular, referee 3 well understood the paper and was especially helpful in making improvements.

Before discussing the referee comments, we would like to address your own views, particularly your statement that “flaws in the assumptions made for the statistical analysis and flaws in providing additional robust evidence to support your conclusion about climatic control in the geological record”.

- 1) We contest the statement that the referees found flaws – or that indeed that there were flaws - in the assumptions of the statistical analysis. First, as concerns the data, we made neither more nor fewer assumptions than the specialist authors that prepared them; we took the series as input data. This is more or less standard practice in paleontology where it is common to analyze data prepared by others. Of course, when interpreting any data, one needs to be aware of its inherent limitations, but here the whole point was to analyze the *actual chronologies* as reported by the specialists. Irrespective of how the chronologies were obtained, we simply quantified their *actual* statistical properties *including correlations that will bias standard interpretations*. Indeed, any second guessing – any attempts to “improve” the empirical chronologies - would have invalidated the conclusions concerning the statistical properties of the raw chronologies.

Our study is little more than a set of empirical analyses taking the data from the cited sources and using them in conventional ways. The originality was in the use of a new paleo indicator: the measurement density. Beyond that, the statistical analyses have been developed in the nonlinear geophysics literature over decades. It is relevant that the lead author has published over 200 papers spanning over four decades, and among other honours, was awarded the Richardson medal of the European Geosciences Union precisely for pioneering work in scaling and scaling analyses of geodata (he is also a fellow of the AGU).

- 2) What may have been interpreted as flaws concerned referees 1 and 3 were the classical problems of interpretation posed by all paleoindicators: the complexity of the underlying processes. Even if the specialists who performed state-of-the-art analyses of their data (including geochronologies) made unwarranted assumptions used flawed methodology, our analyses still characterized the actual series as they are.
- 3) We never claimed that there was any “climate control”. The notion we used is rather one of underlying dynamical regimes (e.g. [Lovejoy, 2023]). These are determined by complex nonlinear geoprocesses acting over wide ranges of scale in space and in time, including (but not limited to), the climate and life (see for example our Nature paper [Spiridonov, A. and S. Lovejoy, 2022: Life rather than climate influences diversity at scales greater than 40 million years. *Nature*, **607**, p 307–312, <https://doi.org/10.1038/s41586-022-04867-y>. Online version. Press release.\). Up until now, these regimes have mostly been identified using paleoclimate indicators, but finding evidence for these regimes in a new paleoindicator - measurement densities - gives support to the idea that the dynamical regimes are indeed more fundamental than simply the climate. In short, we give evidence of dynamical regimes shared between climate indicators and measurement densities, but one does not “control” the other in any way.](https://doi.org/10.1038/s41586-022-04867-y)

Finally, we were puzzled at your last comment: “the need to extrapolate your analyses beyond the Phanerozoic”. Why this is an imperative? The paper already covers 8 orders of magnitude in scale and enlarging the study from the Phanerozoic to cover the largest possible Earth history time scales would increase this by one order of magnitude: to 9. Of course, had data in high enough quantity and quality been available over these very long time scales, we would certainly have done the analyses already! Fortunately, while we are waiting for science to improve its understanding and quantification of these very deep time (Precambrian eon) geological divisions, all our important scientific points and conclusions are adequately covered by the range of several years to several hundred million (Phanerozoic eon).

We hope that the new manuscript is acceptable for publication.

-Shaun Lovejoy (for the authors)

The authors' responses are in italics:

Reviewer #1 (Remarks to the Author):

SUMMARY

The manuscript introduces an important, under-appreciated consequence of the incompleteness of the stratigraphic record: measurement density varies much more notably along geologic timelines than might be anticipated from the sample spacing on a thickness scale. This aspect deserves a wide audience and would be the primary reason to publish this manuscript.

Au: We thank the referee for the enthusiastic endorsement of these first ever analyses of measurement densities!

I recommend returning it for substantial revision, however, because I worry about two potential audiences. Those who struggle with the mathematical foundation of stratigraphic incompleteness are likely to get lost; I know from workshops and courses that the fractal properties easily become confusing. In contrast, readers who are very familiar with the consequences of stratigraphic incompleteness are likely to be frustrated by ambiguities and vagueness in the statements about measurements and scales.

Au: Our statements were simply comments to help the readers understand the quantitative analyses. The analyses were neither ambiguous nor vague, they were obtained using state-of-the-art data analysis techniques. The high level interpretations of the analyses were perhaps simply too concise for the referee? In the new version, we have added an extra figure (fig. 2) with discussion to help make the analyses easier to apprehend.

The manuscript refers to data sets that range from single, local, measured sections to what must be composite sections compiled from multiple local sections.

Au: With the exception of the Phanerozoic series (the Haq sea level and the "Grossman" benthic stack), each series was from a single core.

For the latter, how is the underlying correlation problem overcome? It adds another set of uncertainties and assumptions that seem to have been ignored.

Au: To obtain statistics, we averaged fluctuations over all the disjoint intervals at a given scale. Therefore, especially for long series, at each scale Δt , there could be thousands of fluctuations that were used to obtain robust estimates of statistical moments and correlations (the number of fluctuations at scale Δt is roughly inversely proportional to Δt). The details of the basic techniques are described in more detail in the numerous publications that were cited (especially in [Lovejoy and Schertzer, 2012], [Lovejoy, 2015] (appendix) and [Lovejoy, 2023] (appendix B). The issue of uncertainties was specifically addressed in the methods section of the Nature paper [Spiridonov and Lovejoy, 2022].

The stratigraphic record is a much more reliable indicator of the sequence of events than about their temporal spacing. Even sequence, however, is locally incomplete.

Au: We fully agree, that's why it is so important to statistically quantify the incompleteness, and this as a function of time scale. This paper is therefore able to distinguish - and quantify - the "length Sadler effect" (as distinct from the more usual "resolution Sadler effect" quantified and discussed in our companion paper: [Lovejoy et al., 2025]).

Correlating local sections into a more complete, age-calibrated record is fraught with uncertainties and assumptions. The manuscript does not deal effectively with this aspect of the data the authors have chosen.

Au: Our aim is simply to empirically characterize the measurement densities of standard paleo records derived from a diversity of sources. A proper statistical characterisation is needed in order to overcome the various issues that the referee has pointed out.

STRENGTHS

The manuscript deals in detail with an under-appreciated property of stratigraphic time series. Measurement density of the data encountered in local stratigraphic sections and regional composite sequences is rendered much more variable and intermittent than the apparent regularity of sampling. This problem arises for two reasons. First, the physical record is inevitably highly incomplete on a wide spectrum of scales. Second, the correlation of multiple

sections, which is intended to fill-in the local gaps, introduces more questionable, simplifying assumptions. This manuscript deals with the first of these problems. It suggests statistical remedies that will be new to many readers and the paper could become a valuable reference.

Au: We did not in fact suggest remedies. Before remedies we need systematic, empirical character over a wide range of scales; this was the task of the paper.

WEAKNESSES

The two title words do not provide good guidance for readers to anticipate the contents of the paper. Something like "Multiscale Measurement Densities in Geologic Time Series" might serve better.

Au: Thanks for the suggestion, we opted for: "Time scales gaps and scaling regimes in geochronology measurement Densities" in part because our paper is only marginally oriented towards geology (most of the analyses are at time scales < 1Myr, so have resolutions as short as a year).

The dependence of average rate upon the time span of measurement (the Sadler Effect) is a simple and inevitable mathematical outcome for intermittent and reversible time series like sediment accumulation. Yet, it confounds simplistic approaches to geologic time series. Linear interpolation, for example, will most often be ill-advised when assessing the duration represented by the thickness of an interval of a sequence of sedimentary layers. In my experience, it is not an easy concept for geological audiences to appreciate fully. At very short time scales, intermittency results from the particulate nature of sediment. On the scale of decades to millennia, intermittency and reversibility arise from the dynamics of sediment transport, climate change, and sea-level change. The longer-term accommodation of accumulating sediment depends more on the dynamics of tectonic subsidence. This spectrum of causes is well established in publications and should already be so very familiar to geoscientists that it is not advisable to simplify it.

AU: We did not simplify these processes, we only statistically quantified their properties over wide ranges of time scale, and then we interpreted the statistical analyses. This does not imply simplifications of underlying geological or stratigraphic processes.

For current purposes, the most important data in a sedimentary sequence are those that indicate the age of a layer. When this manuscript refers to data, it fails to indicate whether, or to what extent, the data points are indications of age or duration. These data allow a sequence to be calibrated as a time scale. Such data are critical missing information in Table 1, for example, which lists resolution without a sense of the source of the estimate.

Au: It is true that each series was obtained under diverse conditions, and the chronologies established according to procedures that are standard in the corresponding specialized literatures. These details can be found in the sources cited. In this paper, each geochronology – with its measurement densities - was simply taken as an empirical series at the same level as the paleoindicators themselves. Just as it is common to statistically analyse paleoindicators in this paper, we analyse the measurement density itself as a new paleoindicator. Why is this paleoindicator less worthy of analysis and more problematic than the usual paleoindicators?

When the authors refer to statistics of chronologies, do they somehow know the age of all data points, or do they refer only to dated events. Is there an implicit interpolation of age, in spite of all the warnings about this at the heart of the subject? Is a published age-axis taken at face-value? Is the so-called nominal resolution merely what is written on the cited authors' axes?

Au: Yes, we analyzed the "official" geochronologies (those given in the published cited works). This does not imply that we assumed that they were accurate, they were simply taken as the empirical object of analysis, and were statistically characterized as such.

Does the lack of geological clarity, originate because the native English-speaking author is not a geoscientist and the geologic expertise resides among co-authors who are not native English-speakers. The set of authors is impressive and surely brings all the necessary expertise to this subject.

Au: Lovejoy, the lead author, is indeed a native speaker and contrary to the referee's impression, has published over 200 papers in the geosciences since 1979. Lovejoy is a recipient of the European Geosciences Union Richardson medal (2019) and has been a fellow of the American Geophysical Union since 2016. As for the quality of the writing, referee 3 compliments the authors very highly: "I commend the authors for a nicely written text that was generally a joy to read".

DETAILS

Line 11 . . . knowledge of the Earth's past . . . Delete "the"

Line 11 . . . interpreting successive layers . . .

Line 12 ". . . sheets of rock . . ." I am unsure what is imagined.

Line 25 . . . thus quantitatively explaining the Sadler effect." Since the term was coined, the Sadler Effect has been discussed in more than 160 papers concerned with a wide range of natural processes. Its mathematical background was provided in several papers prior to this, notably by statistician David Strauss, co-authoring with Sadler, but also by others.

Au: In this and in our companion paper [Lovejoy et al., 2025], we show that there are actually two different Sadler effects (the original "resolution" effect and the length Sadler effect), each depending on precise yet different statistical exponents.

Line 33 I agree that temporal resolution is highly variable. How, then, should we understand the single values in the 7th column in table 1? That table appears to give no information about the number of age control points. For speleothems, age control may be provided by counting annual rings but is also provided by Uranium-Thorium systematics. There is also the confounding factor of the relatively narrow range of the heights of the speleothems that have been dated.

Au: The resolutions are nominal, i.e. those given in the source. We did not attempt to modify the estimates given in the cited sources.

Line 46 ". . . if time is discretized at the finest resolution . . ." Sediment is particulate. Would not the finest resolution be the arrival of individual particles? It may be worth considering the limiting time scale of the instantaneous arrival of individual sediment grains. At this limit, the accumulation rate tends to infinity and the stratigraphic completeness to zero. Although this might seem beyond practical value, it is another way to appreciate the inescapable logic of the Sadler Effect.

Au: Clearly, these issues exist and are important, but our outside our scope. We simply used the resolutions that were given in the cited sources and then quantified them statistically.

Line 47 Perhaps there are three options at any location on the time axis: either there is sediment/rock of that age or none and, if there is sediment/rock, there may be a measurement or none. The time and the thickness axes need careful separation. As the paper makes clear, elsewhere, a regular sampling interval in thickness is likely to be quite irregular in time. Reference to thickness and to time scales needs to be more carefully distinguished. Reference to time scales needs to be warily justified.

Au: The problem of thickness versus time scale is indeed fundamental and requires specific modelling, it is outside the scope of this paper. Here, we simply statistically analyze the empirical measurement densities. We could note that some new results on this (the Compound Multifractal Poisson process) have recently been published, and used to model the Geological Time Scale [Lovejoy et al., 2025].

Line 51 Delete "ref"

Line 55-57 Delete "in ref". The manuscript uses three formats for citations: a superscript number alone, a superscript number preceded by "in" and a superscript number preceded by "in ref." The first format dominates and would seem appropriate everywhere.

Line 59 ". . . geochronology measurement densities . . ." Geochronology is the "study of time in relationship to the history of the Earth" (AGI Glossary of Geology). Here, as in the title, it seems simpler and more appropriate to use the word "time series". The current wording leaves a geologist wondering whether all the measurements might be dated events. Given the large numbers of measurements reported, we later wonder whether any of the data points are age calibration events.

Au: According to our team of authors specializing in Holocene, Quaternary and Phanerozoic, the term "geochronology" is not confined to deep time. Indeed, they concur that geochronology means "the science of determining the age of rocks, fossils, and sediments. Its scope is defined by its goal (dating Earth materials) rather than by a specific age range" (Deep Seek). Therefore the term "geochronology" is appropriate for this paper.

Line 70 What is the "10 kyr nominal resolution"? Does "nominal" refer to the original scaling ticks on the age axis? Does it refer to the age bins into which the author compiled data from many sources, with their own uncertainties. Table 1 lists a 6 kyr resolution for this data set, not 10 kyr. The number of data points allows only 1 data point per

approximately 40 kyr, on average, across the 473 Myr time span. Perhaps this is enough that the authors can determine where I became confused.

Au: "Nominal resolution" refers to the resolution of the records attributed by the cited source. We did not attempt to second guess this. It was only used in the analysis to determine the smallest time scale that we analysed. Beyond that, it was reported as an indication of the data type and quality.

Line 73 Do these paleoclimate series comprise proxy data measurements or interpretations?

Au: The paleoclimate series comprise proxy data measurements. The exception is the reported temperatures (the right hand scale in fig. 1) that are based on ¹³C and ¹⁸O isotope measurements as detailed in the cited reference.

Line 76 "We . . . discretized the time axis into bins at the highest (nominal) data resolution." Was this highest resolution only locally realized or was it the highest resolution that could be applied throughout the series. Did it rely on the original authors' scale markings or did it consider the available dated/datable events that that enabled age-calibration. Was interpolation of ages used? How?

Au: We considered the geochronologies - with all their assumptions, difficulties and limitations – to be an empirical data set whose statistics were simply determined over wide ranges of scales. A consequence was that we determined the scale by scale correlations between the measurement densities and the paleo indicators and this information is needed in order to unbiased statistical analyses of the indicators. Although it was out of the scope of this paper, knowledge of the statistics of the chronologies could – and should – be used to improve their quality and interpretations.

Line 94 "H controls the basic scaling of the sedimentation rate." Although not in the form of the Haar function, Sadler's 1981 and later papers all describe incompleteness and patterns of unsteadiness in terms of the values and inflexion points of the slope of logarithmic graphs of average rate against time span.

*Au: Mathematically, geometric sets of points that are scaling are fractal sets, and mathematical functions (such as densities, or paleotemperatures) are multifractals. The two are related as discussed in great detail in the companion EPSL paper [Lovejoy et al., 2025]: "**Appendix B. From fractal sets to multifractal densities: Box, information and correlation fractal dimensions**". In both case, there is a (related) hierarchy of scaling exponents. Typical fractal analyses estimate only one of these exponents, giving only a partial description of the process.*

Line 169-70 The Haq et al. Curve was indeed "assembled" from different outcrops and basins. It should matter to this manuscript how the data were "assembled". Does sequence stratigraphy assume a knowledge of the causes and durations of sequence, their bounding gaps and their "orders," which might compromise the use of the data-set here (line 230).

Au: We made no assumptions beyond those of the cited source. We analyzed and statistically characterized the series as it was.

Line 207 A missing space after "as"?

Au: Thanks.

Line 208 Only long series have the opportunity to span across a long gap, assuming these are gaps in rock/sediment (What geologists often term a hiatus.). This is one of the trivially obvious aspects of the Sadler Effect. For a measured time interval to include a hiatus, that interval must begin and end in rock (or sediment) that can be dated before and after the hiatus; it must be longer than the duration of the hiatus.

Line 220-1 There is a risk that this statement might imply that the more significant and dramatic changes must take more time. The geologic record surely shows that some dramatic extinctions were rather rapid. Of course, in order to appreciate "dramatic" we need a longer record of normal, background rates of change. Of course, longer measurement series improve the chance of capturing "rare" events.

Au: Yes, this was the meaning implied.

Line 248 Double commas; delete one.

Au: Thanks.

Line 251 Column justification fails in columns 5 and 6 of table 1.

Au: Thanks.

Lines 251-2 How does the stated resolving power (7th column in Table 1) compare with the analytical uncertainty in the dating method and reasonable estimates of incompleteness? How many of the data points are determinations of age? The numbers of data points (8th column) rarely amounts to enough for 1 point per bin at the state resolution. What am I missing?

Au: The “resolution” was simply the nominal resolution discussed above. This has been clarified in the table heading.

Line 265-6 This text implies that the data are available on a time axis, that temporal density is known. But the geologic time axes are not known with such resolving power, especially if linear interpolation between dated events is discredited.

Au: Over a given time interval Δt , the calculation of the Haar fluctuations involves summing all the measurements over the interval $\Delta t/2$, and subtracting it from the total over the next interval $\Delta t/2$. It is only the total number of measurements over intervals that matter i.e. their resolutions and exact positions on the time axis do not matter whenever the uncertainties are of shorter duration than $\Delta t/2$. Therefore there may be minor statistical issues at the smallest $\Delta t/2$ but for $\Delta t/2$ much longer than this resolution, they are not important at all (and no interpolations are needed!).

Line 270 Reference to a paper that is in-preparation does not assist the reader. Delete “ref.”

Au: This is the now published EPSL paper: [Lovejoy et al., 2025].

Line 396-297 Gaps result from both non-deposition and erosion. There need not have been “whole swaths of sediment that have completely disappeared from the record.” There are times and places where none were deposited. Neither should the reader be thinking only of large gaps.

Au: Thanks.

Reviewer #2 (Remarks to the Author):

This is the kind of study that many geologists have trouble with: data, including large data sets, being analyzed entirely divorced from their origins. Two observations about statistical analysis are pertinent: 1) It is possible to perform sophisticated statistical analysis data sets as though patterns in numbers alone can reveal significant truths. 2) An elementary requirement for meaningful statistical analysis is to ensure that the sample is representative of the population to be studied. Yes, we are told that the samples come from ice cores, or deep-marine sedimentary records, but there is no context to the individual records. There is no evidence that any attempt has been made here to explore either of the above two points.

Au: Each of the specialist authors provided a representative ensemble of records, totaling 23 records in all. Obviously more could have been added, but systematic statistical analysis revealed much commonality in their statistical structures so that are conclusions are already robust. Although it is true that each record has specific context and for interested readers, these are detailed in the cited sources. Nevertheless, our empirical characterizations of the records using state-of-the-art statistical methods revealed common features. Indeed, it would be surprising if there had been none.

This is the first study of a new paleoindicator: the measurement density, yet there is no reason to think that it is less informative or less reliable than the standard paleoindicators, each of which has various limitations, problems.

The thing about geological data is that it is often very messy. We are talking about the stratigraphic record here, so there are all kinds of reasons why sampling may be incomplete: 1) the sampled item, such as a specific type of fossil, is limited by its ecology to only certain types of rocks, representing specific types of sedimentary environment. The occurrence of these represent at least two uncontrolled variables; 2) the rocks to be sampled are present/absent in response to any kind of geological variable – subsidence or uplift of the sampled basin, climate, selective erosion

reflecting the rocks characteristics, etc. 3) Sampling may simply reflect availability, which is never amenable to regular spaced collection except from continuous drill cores. How many of your source data bases are of this type?

Au: We characterized the actual empirical observations of measurement densities as reported by the specialists. No matter what complexities – and possible errors that were made our - analysis still characterizes the actual existing geochronologies.

We fully realize that the physical processes, the data extraction and chronology estimation are messy; we entrusted this to the specialists and - like many paleoscientists – we analyzed their publicly available published products. Indeed, the issues he/she raises apply to the interpretation of conventional paleoindicators not only to corresponding measurement densities. In our paper, there was no attempt to minimize or ignore any of the numerous issues raised by the referee,

On lines 26-31 the authors acknowledge reasons why geochronologies might be non-uniform because of a range of uncontrolled variables, and then in the next paragraph it seems to be assumed that large data bases then lend themselves to meaningful statistical analysis of this irregularity, proposing that the statistical signals that emerge then have some geological meaning.

Just to take the very large data set used in the construction of Figure 1. Sample density is calculated and displayed graphically as if it has some geological significance. An examination of the paper from which this data set was obtained (reference 29) reveals all kinds of reasons for variable sample density: the data consist of several types of fossil, each constrained by its own ecological parameters; the samples were obtained from both carbonate and phosphatic sediments, which occur in different depositional settings; some of the intervals within which sample spacing is large represent time intervals of different climate or tectonic setting. It is hard to reason why sample spacing, in and of itself, would have any significant meaning at all.

Au: Our paper is an essentially empirical study of a new paleoindicator: the measurement density. The main point of the paper is not to attribute meaning to the measurement density, but rather to quantify it as an empirical characteristic of a paleo record and to show how it directly (via the correlations at each scale) biases the statistics of the classical paleoindicators. As we quantitatively show, this density is often strongly correlated with the indicator itself, thereby biasing the statistics. Why should the measurement density be of any lesser interest than the usual indicators? Indeed - as our correlation results show - understanding the measurement density is needed in order to unbiased the usual paleoindicators.

The thrust of the paper seems to be to explain sample variability entirely as some sort of product of climate change.

Au: This is a misunderstanding: scaling is needed in order to define underlying dynamical regimes but dynamical regimes are presumed to involve numerous interacting and highly nonlinear geoprocesses. As expected, these underlying processes express themselves in both paleoclimate indicators and in measurement densities, yet there is no assumption that climate change - or any other single geoprocess - is a cause it is rather that the measurement densities and the paleoindicators appear to share a common dynamical range.

On line 228 reference is made to a single study (reference 31) that identified the long 2.4 my orbital cycle, and the authors seem to extrapolate from that to a broad assumption of climatic cyclicity everywhere. This particular referenced study (which I looked at), based on solid geological research, might well be correct. The 2.4 my cycle is well known in the geological record. But this is not proof that the entire study under review here and all the data used therein reflect climatic control on different time scales.

Au: Again, there is no assumption of "climate control"; it is rather than the analyses enabled us to identify underlying scaling regimes in the measurement density and these appeared to be close to the regimes found in climate proxies. Dynamical regimes are theorized as having highly nonlinear space-time dynamics such that relevant dynamical processes share a scale invariance symmetry over a range. In a scaling regime, fluctuations vary with scale in a power law manner so that fluctuations over short or long intervals of time differ only in their amplitudes; they are qualitatively the same [Lovejoy, 2023]. The finding that measurement densities and climate proxies share the same scaling regimes is thus expected, but does not imply than one of the processes "causes" or "controls" the other.

In fact the orbital signal is not often clearly preserved in the sedimentary record because its influence is often quite subtle, and is swamped by other non-systematic geological processes, notably tectonic controls and the variability of sedimentary environments and processes. Please see my book "The origin of stratigraphic sequences" for reference.

Au: We relied on the cited sources and accepted any assumptions that they made, we did not make any additional assumptions (e.g. a 2.4 My orbital cycle). In any case, almost all (21 out of 23) of the records analysed were shorter

than 1 Myr.

The best things I can do is suggest that the authors read

Miall, A. D., 2015, Updating uniformitarianism: stratigraphy as just a set of "frozen accidents", in Smith, D. G., Bailey, R., J., Burgess, P., and Fraser, A., eds., *Strata and time*: Geological Society, London, Special Publication 404, p. 11-36.

Au: Thanks! The Miall paper is quite compatible with ours. He acknowledges that:

"The durations of stratigraphic gaps, the distribution of layer thicknesses, and sedimentation rates have fractal-like properties, facilitating the integration of our knowledge of the processes of accommodation generation with data on varying sedimentation rates and the scales of hiatuses and processes operating over all time scales...."

Or later:

"It is now widely recognized that not only the durations of the gaps, but also the distribution of layer thicknesses and sedimentation rates in stratigraphic successions have fractal-like properties..."

Thinking of the strata as "frozen accidents" does not change the fact that the accidents may follow scaling statistics, in agreement with our quantitative analyses and conclusions. Our data analysis makes only the mild assumption that the statistics of the fluctuations are stationary, i.e. that the processes that determine strata, gaps etc. act in the same manner at all the observable times. Statistical stationarity is a type of uniformitarianism.

This paper is attached to my review. The purpose of the 2015 paper was to examine the Sadler effect from the point of view of sedimentology, and to discover what actually drives the geological controls behind rates and styles of stratigraphic preservation at different time scales. Sample spacing was not a variable examined. And there is no statistics in the paper.

Au: This statement is a bit incredible. The entire paper is nothing but statistics! Every single analysis quantifies the statistics of fluctuations in measurement densities and in paleoindicators. There is not a single deterministic analysis anywhere! There seems to be a big gap in the understanding of our methods, motivations and conclusions.

The raw field data speaks for itself. It reveals a range of non-periodic geological processes acting simultaneously over a wide range of time scales. The only statistically regular geological control is astronomical, in the form of orbital forcing and tidal processes, and this is one of the major processes for sedimentary successions that can be dated as being formed within the tens to hundreds of thousands of years time range. Most geological sections are accumulated over periods of millions, or many millions of years, and geological processes acting over those time scales (including long-term climate controls) are episodic or irregular in time and rate.

My suggestion: send this paper to a full-time earth scientist who has complementary skills in statistical analysis.

Au: We are a bit perplexed that the referee suggests sending the paper to "full-time earth scientist who has complementary skills in statistical analysis". All the authors are full-time earth scientists and the lead author has been working in the field of nonlinear geophysics for over forty years and this includes over 200 papers each of which exclusively uses statistical techniques. Indeed, the author has co-invented several new statistical techniques precisely needed for analysing geo data spanning many orders of magnitude in space and/or in time, and this includes the fluctuations analyses used in this paper. This work has recognized by the American Geophysical Union (Lovejoy was elected fellow in 2016), as well as the European Geosciences Union who awarded him the Richardson medal in 2019.

Reviewer #3 (Remarks to the Author):

Review of Scaling Geochronologies

I like the general concept that there is information encoded in the lack of stratigraphic record as it imparts a constraint on the temporal significance of unconformities (or other processes blurring or erasing the record). Basically, this study suggests that measurement density itself, normally considered just a methodological limitation, could act as a new kind of proxy for understanding geological processes. This is a neat idea, and I commend the authors for a nicely written text that was generally a joy to read. I do have some comments which I hope are useful for the authors as they enhance the work. I look forward to seeing this work in print. Well done!

Au: This referee has well understood the basic point of the paper! Thank you.

While we expect that older records have more missing data (this is intuitively self-evident), the key contribution here as I see it is; A statistical model that predicts the growth of missing intervals based on real data. A scaling law that describes how the probability of missing data changes over time. A recognition that correcting these biases when reconstructing past environments may be important.

Au: This is a good summary of the paper.

More significant thoughts.

The concepts here are so important hence it is unfortunate that there are no examples from anything older than the Phanerozoic. I realise this is a moderately large ask but the use of this work would be vastly enhanced with something from the Precambrian. The Phanerozoic is only 12% of Earth's history, and given the work is about sampling density at the grandest conceptual scale it would seem reasonable to have a Precambrian example (of which I can think of plenty to explore).

Au: We totally agree that analyzing series longer than the Phanerozoic would be important. However in order to make convincing analyses, series with at least a hundred or so events are needed, and at the moment this seems to be lacking in the deeper geological record. Nevertheless, using the techniques discussed in this paper, the analysis of the Phanerozoic itself was taken to a new level in [Lovejoy et al., 2025] who analyzed the Geological Time Scale. Since this was composite series, it had no holes, but did display strong clustering that was quantified in the same framework as discussed here.

Many readers would likely appreciate a stronger connection between the conceptual framework and the rock record. Introducing a brief discussion on the diversity of unconformities, perhaps with a specific example or two, would enhance the text and provide greater geological context. A sentence or two in the introduction could effectively ground the discussion in observable field relationships, reinforcing the relevance of these concepts to real-world stratigraphic and tectonic settings.

Au: We're not totally clear what the referee has in mind here. The companion paper studying the properties of the Geological Time Scale may address some of these issues?

Line 77 is important, and it is unlikely that anyone but specialists will get "is the (binary) indicator function of the boundary". Specifically, the concept that the binning is done at the highest possible resolution meaning the smallest available time step in the dataset needs more explicitly discussed. This process effectively transforms the temporal distribution of measurements into a sequence of ones and zeros, much like a presence-absence function. points to consider.

Au: Thanks for the suggestion, we have added a sentence.

Line 12; the word stratigraphy would be useful here as applied to rocks, sediments, and ice. There of course is also the deep time record retained within magmatic systems and their inherited mineral cargo which is also not immune to the same data density concepts.

Au: Thanks, we have added it.

Line 13; insert the word time, so its time gaps.

Au: Thanks, we have added it.

Line 16; I would urge a little caution here as while sedimentary rocks evidently are driven by climate in part, there is also the reasoned counter argument that the rock cycle itself influences atmospheric cycles. Some concept of the interconnection across various temporal scales of the various planetary geochemical reservoirs would be nice to acknowledge.

Au: Thanks, we have added it.

Line 18; what is the difference between measurement density and number of measurements per unit time. If they are the same, as I presume, they are, then no need to repeat this statement as it serves to confuse not clarify.

Au: Thanks, we have clarified this.

Lines 29-40; The Lomb periodogram for unevenly sampled data see Press et al. 1992 probably deserves a mention.

Au: We referenced [Lovejoy and Spiridonov, 2024] where the limitations of the Lomb periodogram when applied to scaling data are discussed with examples from paleoseries.

Lines 39-40; This sentence really needs to be unpacked to help the readership, many of whom will not be time series experts but rather practitioners.

Au: Some changes were made.

Line 49; To be honest the purpose of mentioning meteorological stations in respect to what you are describing is lost on me, surely there is a more relevant example given the papers focus.

Au: At the theoretical level, the problem of the inhomogeneous, scaling measuring stations is the same as in scaling, inhomogeneous time series, the difference being that one is in space, the other in time.

Line 77; I suspect mentioning the classic Haar wavelet, moving window, first before getting to the fluctuation would help the readership.

Au: Thanks!

Line 20; a bit pedantic but for essentially all Earth Science measurements it is five POTENTIALLY independent pieces of information; the paleoindicator, the uncertainty on the paleoindicator measurement, the data density, the time indicator, the uncertainty on the time measurement indicator. While these factors can be analysed separately, they are often interdependent, especially when measurement gaps or dating uncertainties influence how we interpret paleoindicators.

Au: Thanks!

Line 240; “infinite” seems a little of an over sell.

Au: Yes, changed.

It would strengthen the paper to explore potential underlying physical processes that govern the scaling regimes discussed, as this could provide deeper insights into why measurement densities follow the observed patterns. Additionally, I find the argument compelling that assuming missing data is random leads to biased interpretations, and that recognizing a power-law distribution in missing data allows for statistical adjustments in reconstructions. It would be valuable to elaborate further on how such adjustments could be practically implemented (whether through resampling techniques, weighting schemes, or other statistical corrections) to guide future applications of this approach in paleoclimate and geochronological studies.

Au: We totally agree, yet this is the subject of a new paper, new set of models! This paper is restricted in scope to empirical characterizations, although the companion paper [Lovejoy et al., 2025] also introduces the new Compound Multifractal Poisson process as a model for the geological time scale. Consequences need development and this will happen elsewhere.

Chris Kirkland, Perth, WA

References

- Lovejoy, S., A voyage through scales, a missing quadrillion and why the climate is not what you expect, *Climate Dyn.*, 44, 3187-3210 doi: 10.1007/s00382-014-2324-0, 2015.
- Lovejoy, S., Scaling, dynamical regimes and stratification: How long does weather last? How big is a cloud? , *Nonlinear Processes in Geophysics* 30, 311–374 doi: <https://doi.org/10.5194/npg-30-311-2023>, 2023.
- Lovejoy, S., and Schertzer, D., Haar wavelets, fluctuations and structure functions: convenient choices for geophysics, *Nonlinear Proc. Geophys.* , 19, 1-14 doi: 10.5194/npg-19-1-2012, 2012.
- Lovejoy, S., and Spiridonov, A., The Fractional Macroevolution Model, a simple quantitative macroevolution model, *Paleobiology* 2024:1-25. doi:10.1017/pab.2023.38, 2024.

Lovejoy, S., Spiridonov, A., Davies, R., Hebert, R., and Lambert, F., From Eons to Epochs: multifractal Geological Time and the Compound Multifractal - Poisson Process, *Earth and Plan. Sci. Lett.*, <https://doi.org/10.1016/j.epsl.2025.119460>, 2025.

Spiridonov, A., and Lovejoy, S., Life rather than climate influences diversity at scales greater than 40 million years, *Nature*, 607, 307–312 doi: 10.1038/s41586-022-04867-y, 2022.

REVIEWER COMMENTS and responses in italics:

Reviewer #1 (Remarks to the Author):

I remain very much in favor of accepting this manuscript for publication. The statistical Haar-fluctuation method that it presents deserves to be much better known among geologists and paleontologists. I look forward to using that approach and this paper would be my cited source.

The authors' revisions have added considerable clarity to the prior draft and title. I agree with them that little would be added by extending the data deeper into Precambrian time, the multi-fractal nature of the record is well-known.

Au: We thank the referee for this strong appreciation!

I sympathize with the authors' evident frustration at some naivety that the earlier draft exposed in the reviewers' understanding, including mine. We reviewers are, however, likely to have brought more expertise to the topic than the average reader. The author's revisions are surely worthwhile for the overall impact of the publication, even at the cost of lengthening the paper.

I do remain somewhat confused by the intrinsic "nominal resolution" the authors ascribe to their data sources. I would not insist on any revision for this, because understanding their method does not seem to hinge on this value. Perhaps it would be useful, however, if I give an example that might explain my confusion. Consider Figure 12.5 (chapter 12, figure 5) in Elsevier's "A Geologic Time Scale 2004" edited by Gradstein, Ogg and Smith. The figure displays the interpolation of an age-scale through 22 radio-isotopically dated stratigraphic levels into an ordinal sequence of early Paleozoic events spanning almost 80 million years.

The spacing of the 22 dated levels displays the kind of irregularity (whether on the age scale or the composite, ordinal event scale) that the current manuscript will help geologists manage. The resolution of the individual dates is +/- 1.5 to 1.7 million years. The time-ticks on the age axis are 2 million years apart. On the other axis, the 22 dated events are imbedded in an ordinal sequence of the first and last appearance events of 669 fossil graptolite species and subspecies, together with other events (e.g. isotopic anomaly boundaries), for a total of 1,400 event levels. This ordinal composite sequence of events had been optimized, using simulated annealing, from the local records of 119 stratigraphic sections, each with its own unique pattern of gaps and missing events.

By the GTS 2020 edition of the Geologic Time Scale book (this one edited by Gradstein, Ogg, Schmitz and Ogg) the composite ordinal scale had grown to be based on 2,600 Ordovician to Devonian graptolite species range-end events, culled from 840 local stratigraphic sections for a total of 34,000 local events correlated and sequenced.

A second composite sequence for 2,200 Cambrian to Silurian conodont species known from 1,300 sections was based upon a total of 41,000 local event records. Only about 150 stratigraphic sections in this global database yielded both graptolite and conodont species ranges for about 7,500 local range-end records.

Which of all the counts in the previous two paragraphs would be the intrinsic nominal resolution for these data? Although I am unsure, I doubt that has undermined my grasp of the Haar statistics.

Au: Thank you for this helpful information. In our original submission, 21 out of 23 geochronologies that were analysed were from single cores covering most of the Holocene and Quaternary. Of the exceptions - the Haq paleo sea level and Grossman and Joachimski¹ 2022 paleotemperatures - the latter at least actually had – as claimed by the authors of that series - fairly uniform 10 kyr resolutions. In the revised supplementary material (fig. S2) this is shown explicitly. However, thanks to the encouragement of referees and the editor, we found a Precambrian series, Isson and Rauzi 2024²) that we decided to include in order to extend our study further back in time and to longer time intervals. The results are shown below and in the revised text (see the new fig. 3 and the new part 1 of the supplementary material). This chronology is probably the main one to which the complications which you have detailed fully apply. As discussed in part 1 of the SUPP, the variations in the nominal resolutions over the length of the chronology are extreme: 100 years for the youngest parts to 1 Myr for the Precambrian parts (fig. S2). This required extra analyses – including separating the Phanerozoic and Precambrian segments of the data – in order to yield a more convincing result.

*Also in the SUPP, we added this explanation:
and the following discussion was given:*

“It should be underlined that the Haar fluctuations work by first averaging and then differencing so that they effectively systematically degrade the resolutions so that the nominal resolution is really just the highest resolution (smallest Δt) that can be meaningfully used. If the nominal resolution is artificially short, as long as it is more or less fixed (constant) throughout the length of the chronology, it is sufficient to simply ignore the statistics from the shortest Δt . The real difficulty comes when the nominal resolution is highly variable over the length of the chronology. This is case in the Precambrian Isson chronology, which is very short for young sections of the chronology and much longer for the older sections. It turns out that Isson is the only chronology that we analyzed where this is a problem, we investigate it below.”

The Precambrian figure added to the ms. Other expository figures and material were added to the SUPP

Fig. 3: A comparison of the root mean square (RMS) fluctuations of the Isson² $\delta^{18}\text{O}$ in Carbonates (red, roughly calibrated in K) with the normalized measurement density (blue) and the correlation between the two ($R(\Delta t)$, brown, linear vertical scale at right). There were 25399 data points, the oldest being 3.504 Gyrs so that this fluctuation analysis takes us close to the origin of the Earth. As described in the text, the Isson $\delta^{18}\text{O}$ were roughly calibrated so that the results are expressed in units of K. The density fluctuations are dimensionless as is R . The dashed reference lines with slopes 0.3 and -0.05 are close to the RMS temperature fluctuations and normalized density fluctuations and covers the megacimate (≈ 0.5 Myrs to ≈ 0.5 Gyrs).

Reviewer #2 (Remarks to the Author):

I see no discernible response to my comments on the first version of this paper. I attempt to contact Dr. Lovejoy to discuss my issues informally, but have not had a response. As I explained in that communication, with my reasons, I cannot convince myself that geochronology sample spacing has any other meaning than the local availability of sample material. Accordingly I find the subject of the ms misleading, and cannot recommend its publication.

Au: We could make an additional clarification about the Haar fluctuation analysis. In the above, the main point about statistical ensembles is that we could treat each chronology as a member of an ensemble of chronologies and then draw conclusions from commonalities within the ensemble of chronologies. However, in the paper, we first applied the fluctuation analysis to single chronologies. In this case, at time scale Δt , we have an ensemble created by dividing each chronology into disjoint intervals of length Δt : we treat all such intervals as coming from a statistical ensemble. As long as there are not strong variations in the interval statistics over the length of the chronology (e.g. when there is effectively more than one ensemble) this may be a reasonable assumption/approximation. In the 23 series discussed in the original text, this seems reasonable: even as discussed in the response to referee 1, the Grossman and Joachimsky benthic stack seems to maintain a fairly constant nominal resolution over its entire length, see the new fig. S2. The main challenge was the newly introduced Isson and Rausky Precambrian chronology, see the discussion in response to referee 1 (and part 1 of the revised SUPP).

Reviewer #3 (Remarks to the Author):

I remain positive about this contribution, as I was in the previous round. The following minor but constructive points may help the authors further strengthen the paper.

Au: Thanks for the appreciation!

Minor points:

“Much of our knowledge of the Earth’s past is gleaned by interpreting varying stratigraphy of layers in outcrops, cores drilled through lake and ocean sediments, and sheets of rock and ice.” For a geologist this statement is technically a bit limited, in that layers in outcrop implies a metasedimentary pile what about all the information from the igneous rock record? I don’t disagree with the statement at all, just it really would be more accurate for the subject to be framed as “Much of our knowledge of Earth’s past is gleaned from interpreting the stratigraphy of sedimentary, igneous, and metamorphic sequences, whether exposed in outcrop, recovered from cores through lake and ocean sediments, or preserved in crystalline crust.”

Au: OK, thanks, we have used your formulation.

There is a structural issue in logic and temporal framing around line 14. The first sentence sets up a deep-time or all-Earth-history perspective, while the second abruptly narrows to Holocene–Quaternary dynamics without signalling the temporal zoom. To resolve this, the bridge between the two should explicitly define that the case studies or processes examined are focused on the recent geological past, even though the framework could apply more broadly.

Au: Point well taken. It is true that in the original submission only 2 out of 23 chronologies that were analyzed were less than 1 Myr old, and this coloured the framing. Now, with the encouragement of referees and the editor, we have included a newly published Precambrian series, the third in this paper that is squarely in geological deep time. We have added comments in various places to attempt to redress the framing.

Line 33 and 34 “the dated geochronologies are typically non-uniform”, agree with the concept but the word geochronology is a tautology. The phrase “geochronology is typically non-uniform” does the job and consistent with community usage of what geochronology is. By definition, geochronology already refers to the dated temporal framework of geological materials or events. Adding “dated” is redundant.

Au: OK, thanks, we made the change.

Line 37; Personally, I don’t massively like the zoom in to cores specifically, this work is equally as relevant to outcrops, mineral grains or mass spec analyses. As a

geochronologist this framing may lose some readership in my community. I mention this only to be helpful.

Au: Again point well taken, changes made.

Line 46; nice!

Au: thanks!

Line 63-65; Not sure you really need this claim of precedence?

Au: Removed as suggested.

Line 180; “Our knowledge of the evolution of life and the planet is largely inferred from stratigraphic records that are marred by hugely varying sedimentation rates and gaps where swaths of data are missing” Debatable, this overstates the role of stratigraphic records in reconstructing the entire history of life and the planet. Beyond the Phanerozoic, the sedimentary archive becomes increasingly incomplete, while the igneous and metamorphic record, particularly through mineral chronometers and geochemical proxies, becomes the dominant repository of information.

Au: Point well taken. Of course the entire paper was within the Phanerozoic (and only now in the revised version was it extended to the Precambrian), but your point is nevertheless valid. Without wanted to go into a detailed discussion, we softened the sentence to read: “Much of our knowledge....”.

Line 221; Please unpack this sentence a bit more “Our Haar fluctuation analysis directly handles inhomogeneous data (for periodic structures see ref), and has implications for statistical analyses. For example, H is the highest order of differentiation, typically <1 ; therefore, linear (or polynomial) interpolations ($H \geq 1$) will be biased”. This would benefit from being unpacked into two or three sentences that clarify both what the Haar fluctuation analysis does and why it matters for interpolation or scaling.

Au: OK, we have added several sentences of explanation.

Line 231-233; this is probably more standard phrasing for the community; “the gaps in paleo-proxy records become more pronounced at the longest time scales, where the most significant and dramatic changes in biota and the physical Earth become the most apparent.”

Au: Thanks, we have used your proposed wording.

I will highlight in this response not for something here, but maybe in the future. “The concepts here are so important hence it is unfortunate that there are no examples from anything older than the Phanerozoic. I realise this is a moderately large ask but the use of this work would be vastly enhanced with something from the Precambrian. The

Phanerozoic is only 12% of Earth's history, and given the work is about sampling density at the grandest conceptual scale it would seem reasonable to have a Precambrian example (of which I can think of plenty to explore). Authors: "We totally agree that analyzing series longer than the Phanerozoic would be important. However in order to make convincing analyses, series with at least a hundred or so events are needed, and at the moment this seems to be lacking in the deeper geological record" Reviewers comment: It is not lacking in the deeper geological record. May be reach out to me, there are tons of truly deep time records that are sufficiently rich to try this type of approach on.

Au: Thanks, as you can now see, the revised ms does include the analysis of a truly long, Precambrian series. Also, as discussed in the new part 1 of the SUPP, it requires more care than the more straightforward younger series analysed in the earlier versions of the text.

In future, we are indeed interested in analyzing other long data sets and are open to future collaborations. Thanks for the encouragement!

I'll also reiterate my earlier point, which may be helpful here to consider for a final time. As implied by Reviewer 2, broadening the paper's accessibility to a wider geological readership, not only those with a strong statistical background, would be achieved by providing a little more grounding in the rock record. While I appreciate that this is not the central focus of the study, including a few key statements about what the "missing record" looks like geologically would strengthen the conceptual foundation of the work, as this is its most fundamental premise. Even a short addition to the Introduction could effectively ground the discussion in observable field relationships, reinforcing the relevance of these concepts to real-world stratigraphic and tectonic settings. This could be achieved for example, through a few sentences on diverse unconformity types, aiming to frame observable field relationships to the statistical concept, enhancing accessibility for the widest geological audience. This is a strong and interesting paper, and I offer these comments in the hope that they help strengthen its impact.

Sincerely, Chris Kirkland

Au: We've added to the Introduction the following geological-stratigraphical context on the significance of the study framing it in commonly used standard terms of the physical stratigraphy:

"The science of stratigraphy recognized for a long time the significance of temporal gaps in diverse records and across time scales. In concert with the lithological and facies succession features, the genetic discontinuities between neighbouring rock formations are used in identifying the presence of gaps in stratigraphical records, and thus defining sequences in a correlational framework of the sequence stratigraphy which searches the best ways to objectively distinguish space-time 'packages' of the geological record³. A precursor to sequence stratigraphy – allostratigraphy - was and still is explicitly concerned with revealing the ranges of discontinuities and gaps in the geological record, which later

are used in correlation between distant areas ⁴. Gaps appear at all time scales, and depending on their magnitude can be graded from diastems (relatively short interruptions on geological time scales) to disconformities in relatively continuous and lithologically homogenous successions, to angular unconformities which signify not only gaps in the record but also structural and tectonic deformations which happened in the unobserved time, and to complete nonconformities when sedimentary rocks or sediments are found to be overlaying much older igneous and metamorphic rocks, which usually signal gaps with durations reaching 100s Myrs to Gyrs, Therefore the classical stratigraphy already implicitly recognized the scaling and scale free nature of the gaps in the record and chronologies. The present study quantifies this structure of the gaps as a function of time scales over which they occur while also exploring their correlations with proxies, which show clear scale-dependence of these correlations. The presence of correlations between proxies and the gaps in their records suggests two things: i) proxy record is systematically biased in non-trivial ways with the scale-dependent magnitudes of distortion; ii) there should be a common cause, implicit in both allo- and sequence stratigraphical approaches, which drives paleoclimatic signals and the formation of gaps in their records. Therefore, paraphrasing the famous aphorism: “the absence of evidence is also the evidence of absence”. It means that the time scale dependent (scaling) structure of gaps in records of proxies can be used as a new and mathematically tractable source of information in revealing the Earth system processes. “

** Visit Nature Portfolio's author and referees' website at www.nature.com/authors for information about policies, services and author benefits**

- 1 Grossman, E. L. & Joachimski, M. M. Ocean temperatures through the Phanerozoic reassessed. *Scientific Reports* **12**, 8938 (2022).
<https://doi.org/https://doi.org/10.1038/s41598-022-11493-1>
- 2 Isson, T. & Rauzi, S. Oxygen isotope ensemble reveals Earth's seawater, temperature, and carbon cycle history *Science* **383**, 666–670 (2024).
- 3 Miall, A. D. *The geology of stratigraphic sequences*. 2nd edn, (Springer., 2010).
- 4 Catuneanu, O. *Principles of sequence stratigraphy*. (Elsevier, 2022).